# Multiple neuronal networks coordinate *Hydra* mechanosensory behavior

**Krishna N Badhiwala[1], Abby S Primack[2], Celina E Juliano[2], Jacob T Robinson[1,3,4]***

[1]Department of Bioengineering, Rice University, Houston, United States; [2]Department of Molecular and Cellular Biology, University of California, Davis, United States; [3]Department of Electrical and Computer Engineering, Rice University, Houston, United States; [4]Department of Neuroscience, Baylor College of Medicine, Houston, United States

**Abstract** *Hydra vulgaris* is an emerging model organism for neuroscience due to its small size, transparency, genetic tractability, and regenerative nervous system; however, fundamental properties of its sensorimotor behaviors remain unknown. Here, we use microfluidic devices combined with fluorescent calcium imaging and surgical resectioning to study how the diffuse nervous system coordinates *Hydra*'s mechanosensory response. Mechanical stimuli cause animals to contract, and we find this response relies on at least two distinct networks of neurons in the oral and aboral regions of the animal. Different activity patterns arise in these networks depending on whether the animal is contracting spontaneously or contracting in response to mechanical stimulation. Together, these findings improve our understanding of how *Hydra*'s diffuse nervous system coordinates sensorimotor behaviors. These insights help reveal how sensory information is processed in an animal with a diffuse, radially symmetric neural architecture unlike the dense, bilaterally symmetric nervous systems found in most model organisms.

## Introduction

Discovering the fundamental principles of neural activity and behaviors requires studying the nervous systems of diverse organisms. Animals have evolved different neural structures like the nerve net (e. g., *Hydra*), nerve cords and ganglia (e.g., *Caenorhabditis elegans*, *Aplysia*, planaria), and brain (e.g., *Drosophila*, zebrafish, rodents, and primates). Despite the vastly different structures, many behaviors are conserved across species, including sensorimotor responses (*Clark et al., 2013*; *Ghosh et al., 2017*; *Kaplan et al., 2018*; *Ahrens et al., 2012*; *Chen et al., 2018*; *Haesemeyer et al., 2018*) and sleep (*Hill et al., 2014*; *Raizen et al., 2008*; *Artiushin and Sehgal, 2017*; *Guo et al., 2018*; *Kayser and Biron, 2016*; *Gandhi et al., 2015*; *Zhdanova et al., 2001*; *Zimmerman et al., 2008*; *Kanaya et al., 2020*). By comparing neural circuits that support similar behaviors despite different architectures, we can discover organizational principles of neural circuits that reflect millions of years of evolutionary pressure. While there are many potential organisms that would support this type of comparative neuroscience, only a small group of animals have the qualities to support laboratory experiments: short generation span, ease of breeding and manipulation in laboratory conditions, small and compact size, optical transparency, and a well-developed genetic toolkit with a complete spatial and molecular map of the nervous system.

Transparent, millimeter-sized animals in particular offer a number of advantages for neuroscientists because it is possible to image neural activity throughout the entire nervous system using genetically encoded calcium or voltage-sensitive fluorescent proteins (*Broussard et al., 2014*; *Chen et al., 2013*; *Lemon et al., 2015*; *Prevedel et al., 2014*; *Ahrens et al., 2013*; *Cong et al., 2017*; *Kim et al., 2017*; *Portugues et al., 2014*; *Vladimirov et al., 2014*; *Gonzales et al., 2020*). In addition, some millimeter-sized animals are compatible with microfluidic devices for precise

*For correspondence: jtrobinson@rice.edu

**Competing interests:** The authors declare that no competing interests exist.

environmental control and microscopy techniques that offer cellular-resolution functional imaging of the entire nervous system. These properties, combined with genetic tractability, provide a powerful way of revealing neuronal dynamics across the entire nervous system (not just a small region) during behaviors. For instance, whole nervous system imaging of confined or freely moving animals has revealed the neuronal dynamics underlying locomotion (*Prevedel et al., 2014*; *Nguyen et al., 2016*) and sensory-motivated global state transitions in *C. elegans* (*Nichols et al., 2017*; *Gonzales et al., 2019*), responses to noxious odor and visuomotor behaviors in zebrafish (*Prevedel et al., 2014*; *Ahrens et al., 2013*; *Cong et al., 2017*; *Kim et al., 2017*; *Portugues et al., 2014*; *Vladimirov et al., 2014*), responses to light and odor in *Drosophila* (*Lemon et al., 2015*; *Aimon et al., 2019*), and neuronal ensembles correlated with basal behaviors and response to light and heat in *Hydra* (*Dupre and Yuste, 2017*; *Badhiwala et al., 2018*).

*Hydra* is unique among the small, transparent organisms discussed above due to its regenerative ability and highly dynamic nervous system. While most small, transparent model systems (like *C. elegans* or zebrafish larvae) suffer permanent behavioral deficits from the loss of one or a few neurons (*Bargmann and Horvitz, 1991*; *Bargmann and Avery, 1995*; *El Bejjani and Hammarlund, 2012*; *Kroehne et al., 2011*; *Hecker et al., 2020*), *Hydra* can completely recover from a significant neuronal loss to regain normal contractile behavior in as little as ~48 hr (*Itayama and Sawada, 1995*; *Soriano et al., 2009*; *Gierer et al., 1972*). This radially symmetric freshwater cnidarian has a nervous system composed of two diffuse networks of neurons, one embedded in the endoderm and another embedded in the ectoderm (*Burnett and Diehl, 1964*; *Lfntz and Barrnett, 1965*). While *Hydra's* diffuse nerve net is highly dynamic with continuous cellular turnover and migration (*Bode et al., 1988*; *Campbell, 1967*), regions with increased neuron density resembling nerve rings have a comparatively lower neuronal turnover (*Figure 1a*; *Bode et al., 1973*; *Epp and Tardent, 1978*; *Koizumi et al., 1992*; *Hufnagel and Kass-Simon, 2016*). One of these regions is in the oral end in the apex above the ring of tentacles ('hypostomal nerve ring'), and another is in the aboral end in the foot ('peduncle nerve ring') (*Figure 1a*). Recent single-cell RNA sequencing has provided a

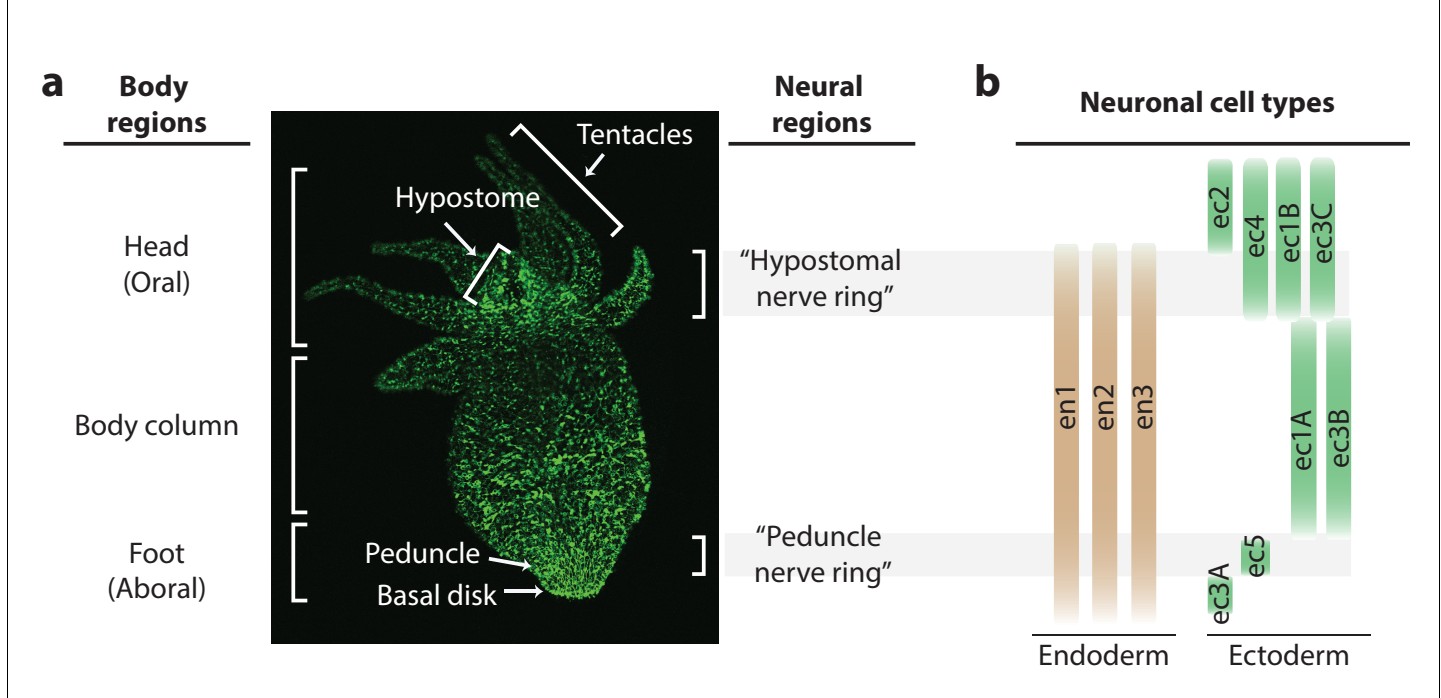

**Figure 1.** Distribution of neurons in the *Hydra* nerve net. (a) Fluorescent image of *Hydra* nervous system. Green fluorescent protein (GFP) is expressed in neurons and neuronal progenitors (nGreen transgenic line; *Siebert et al., 2019*). Body anatomy is annotated on the left. White arrows indicate the body parts: hypostome, tentacles, peduncle, and basal disk. High neuronal density regions are annotated on the right. (b) Distribution of neuronal cell types varying longitudinally along the body, with endodermal nerve net cell types in tan and ectodermal nerve net cell types in green. Cell types were identified through single-cell RNA sequencing (*Siebert et al., 2019*).

complete molecular and spatial map of the *Hydra* nervous system, including identification of unique cell-type-specific biomarkers to generate new transgenic models (*Siebert et al., 2019*). This existing molecular and spatial map of the nervous system suggests that there is no overlap in the neuronal cell types that make up the hypostomal and peduncle nerve rings, and the distribution of the cell types varies along the length of the body (*Figure 1b*). Finally, the demonstration of microfluidic and transgenic tools combined with *Hydra's* dynamic yet 'simple' neural architecture has enabled observations of basal and sensory motivated behaviors in the regenerating nervous system (*Badhiwala et al., 2018*).

To better establish *Hydra* as a model organism for comparative neuroscience, it is critical to understand their basic sensorimotor behaviors, such as response to touch. While it is well documented that *Hydra* contract when mechanically agitated or poked with a pipette (*Mast, 1903*; *Wagner, 1905*; *Rushforth, 1965*; *Rushforth et al., 1963*; *Rushforth and Burke, 1971*), we found no quantitative reports of how this behavior depends on stimulus intensity or is mediated by neural activity. Although significant insights in *Hydra* behavior have been made over the last several decades with simple methodologies and manual observations, including that the tentacles and/or the hypostome are needed for mechanosensory response, these experiments lack quantitative characterization of neuronal or behavioral response. Forceps-induced touch allows local stimulation, but it is difficult to control the force applied manually (*Campbell et al., 1976*; *Takaku et al., 2014*). While stimulation with mechanical agitation allows control over stimulus intensity, these observations are limited to changes in body lengths (*Rushforth et al., 1963*).

Here, we use whole-animal functional imaging combined with resection studies to discover that despite the apparently diffuse nerve net in *Hydra*, these animals process sensorimotor responses in specialized regional networks. To study the mechanosensory response, we first developed a microfluidic system to apply a local mechanical stimulus and quantify *Hydra's* behavioral and neural response. We then measured these responses in the absence of select regions of the body and found at least one of the neuron-rich regions, the hypostome (oral) or the peduncle (aboral), is required to coordinate spontaneous contractions, though the oral network plays a more significant role. We found a significant reduction in the mechanosensory response with the removal of the hypostome, the region where sensory information is likely processed. These sensorimotor experiments combined with whole-animal neural and epitheliomuscular imaging reveal that *Hydra* is capable of receiving sensory information along the body column; however, the oral region is necessary for coordinating the motor response.

## Results

### *Hydra's* mechanosensory response is dependent on stimulus intensity

To better understand sensory information processing in *Hydra*, we developed a double-layer microfluidic system that can apply a local mechanical stimulus while we image the response of the entire nervous system using fluorescence microscopy (*Figure 2*, *Video 1*). This local mechanical stimulation is made possible by push-down microfluidic valves (*Figure 2a*) that deliver mechanical stimuli to a portion of the *Hydra* body with precise temporal and spatial control (see Materials and methods). For all experiments, we pressurized a valve (400 µm diameter) that was directly above the animal (for 1 s every 31 s, see Materials and methods) to stimulate the body column while simultaneously performing functional calcium imaging (*Figure 2b*). We selected the middle of the body for stimulation region to help ensure that we stimulated roughly the same region of the animal throughout each experiment. This choice was based on the observation that the body column region was relatively stationary, whereas the oral and aboral extremities had large displacements during body contractions and elongations.

Experiments showed that this stimulation paradigm delivered a local mechanical stimulation with most of the mechanical force localized to a radius of approximately 250 µm around the microfluidic valve. To measure the locality of this stimulus, we performed an experiment using transgenic *Hydra* (nGreen) (*Siebert et al., 2019*) expressing GFP pan-neuronally (and in neural progenitors) and tracked the position and fluorescence intensity (GFP) from individual neurons during mechanical stimulation (N = 222 neurons over 1 min). When the microfluidic valve was pressurized to deliver mechanical stimulation, we found significantly increased average cellular (or tissue) displacement

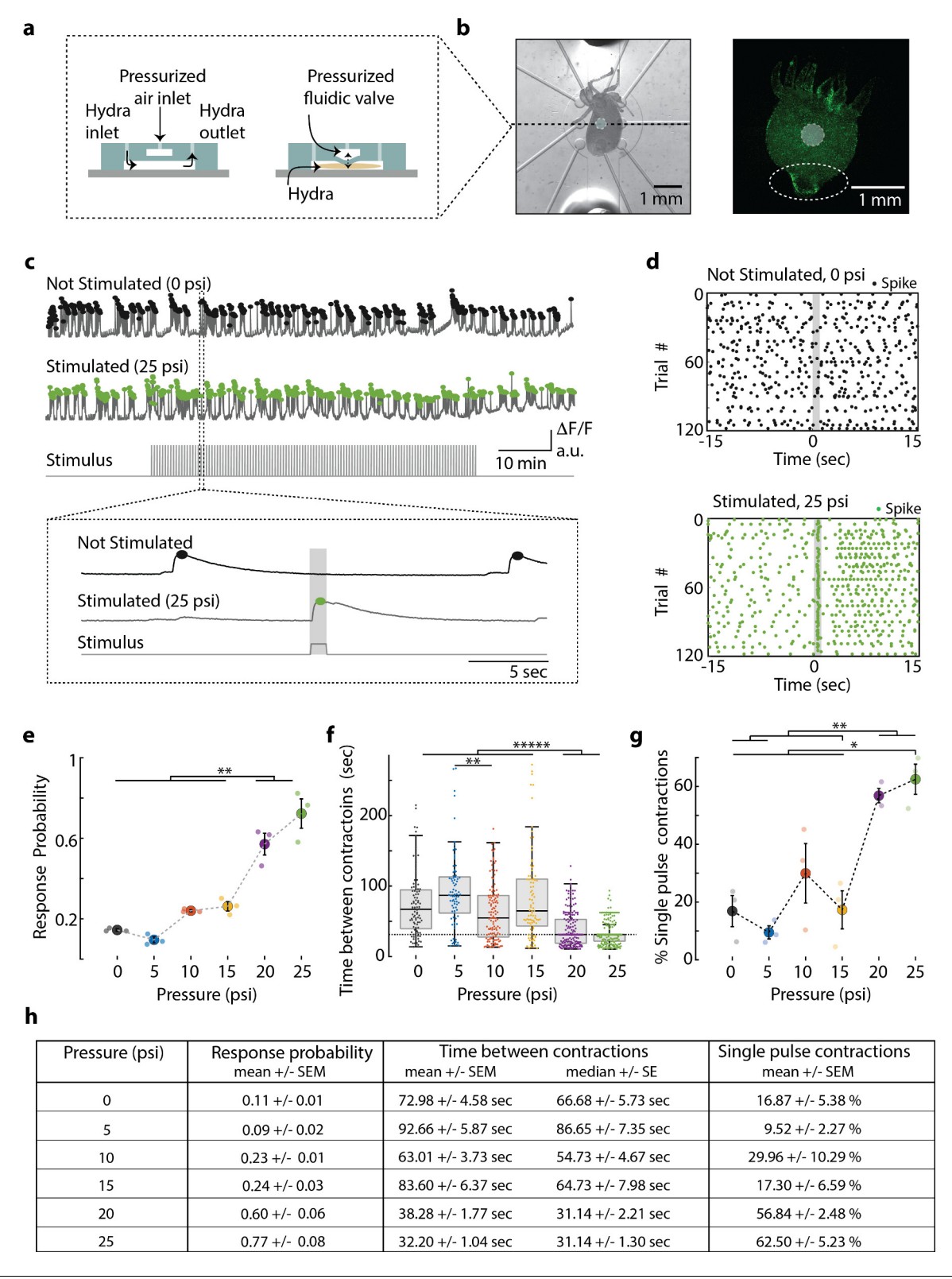

**Figure 2.** *Hydra's* neuronal response depends on the mechanical stimulus intensity. (a) (Left) Side view of double-layer microfluidic device for mechanical stimulation. (Right) Device with pressurized valve. *Hydra* is immobilized in the bottom *Hydra* layer, and pressurized air supplied into the valve layer causes the circular membrane (400 μm diameter) to push down on *Hydra*. (b) (Left) Brightfield image of *Hydra* immobilized in the bottom layer of the chip and the arrangement of micro-valves on the top layer. The micro-valve used for stimulation is falsely colored with a light blue circle.

*Figure 2 continued on next page*

*Figure 2 continued*

(Right) Fluorescent image of *Hydra* with pan-neuronal expression of GCaMP6s. White dashed circle marks the peduncle region of interest (ROI) used for quantifying calcium fluorescence changes. (**c, d**) Representative calcium fluorescence activity in the peduncle region from an animal not stimulated and an animal mechanically stimulated with 25 psi. Black and green dots indicate fluorescence (calcium) spikes. Gray shaded regions indicate stimulus 'on' time (also 'response window'). (**c**) Fluorescence (calcium) trace from *Hydra* not stimulated (top) and stimulated with 25 psi (middle). Stimulation protocol in gray (bottom trace): 20 min no stimulation, 1 hr of repeated stimulation (1 s 'on,' 30 s 'off') and 20 min no stimulation. Stimulus 'on' times indicated with vertical lines. Magnification of 30 s fluorescence and stimulation protocol trace from one stimulation trial. (**d**) Raster plot of stimulus time-aligned spiking activity from multiple trials superimposed for *Hydra* not stimulated (top) and stimulated with 25 psi (bottom). (**e**) Mechanosensory response probability, fraction of trials (out of 119 total) that have at least one calcium spike (also contraction pulse) occurring during the 1 s response window (gray shaded region) when valve is pressurized. Large circles indicate average probability from all animals (N = 3) combined for each condition. Small circles indicate probability from a single animal. Significant pairwise comparisons are shown with brackets (one-way ANOVA with post-hoc Bonferroni correction). (**f**) Time interval between body contractions under each condition. Dashed line represents the time interval between stimuli (~31 s). Brackets indicate significant differences in a Kruskal–Wallis test with post-hoc Dunn–Sidak correction. (**g**) Percent of all body contractions that are a single pulse; brackets show significant pairwise comparisons from a one-way ANOVA with post-hoc Bonferroni correction. Error bars are standard error of mean (SEM); N = 3 *Hydra* for each condition; *p<0.05, **p<0.01, *****p<0.00001. (**h**) Table summarizing the mechanosensory response probability, time between contractions, and percent of contractions that are single pulses for each stimulus intensity (mean ± SEM or median ± SE). Source data for the quantitative characterization of mechanosensory response are available in *Figure 2—source data 1*.

The online version of this article includes the following source data and figure supplement(s) for figure 2:

**Source data 1.** source file for stimulus-dependent mechanosensory response.
**Figure supplement 1.** Distribution of mechanical forces.
**Figure supplement 2.** Average calcium fluorescence from large peduncle region of interest (ROI) correlated with calcium fluorescence from smaller ROIs for individual peduncle neurons.
**Figure supplement 3.** Spikes in calcium fluorescence are due to calcium activity not motion artifacts.
**Figure supplement 4.** Simultaneous electrophysiology and calcium imaging of ectodermal epitheliomuscular cells.
**Figure supplement 5.** *Hydra*'s epitheliomuscular response is dependent on the mechanical stimulus intensity.
**Figure supplement 6.** Mechanosensory response window.
**Figure supplement 7.** Mechanical sensitivity of different body regions in *Hydra*.
**Figure supplement 8.** *Hydra*'s mechanosensory response time is faster than passive calcium diffusion through epitheliomuscular cells.
**Figure supplement 9.** Long-term mechanical stimulation.

(p<0.001). Further analysis of the cellular movements showed the spatial distribution of mechanical force from the stimulation was primarily experienced by the neurons directly under the valve (*Figure 2—figure supplement 1*). Specifically, we found that the tissue directly below the valve was compressed (z direction) when mechanically stimulated – the neurons directly under the valve had a small magnitude of lateral (x–y direction) displacement. The tissue bordering the valve was stretched away from the valve center – the neurons in the neighboring regions around the valve had the largest lateral displacement. This lateral displacement decreased for neurons that were farther from the center of the valve. Neurons more than 750 μm from the microfluidic valve center showed a negligible displacement of less than 5 μm (95% CI lower bound = 5.8 μm), which is ~550% less than the displacement of neurons bordering the valve.

Having established our method to provide local mechanical stimuli, we characterized *Hydra*'s sensitivity to local touch and the associated neural response. We performed experiments using transgenic *Hydra* expressing GCaMP6s in neurons (*Dupre and Yuste, 2017*). When we delivered mechanical stimuli, we found bright

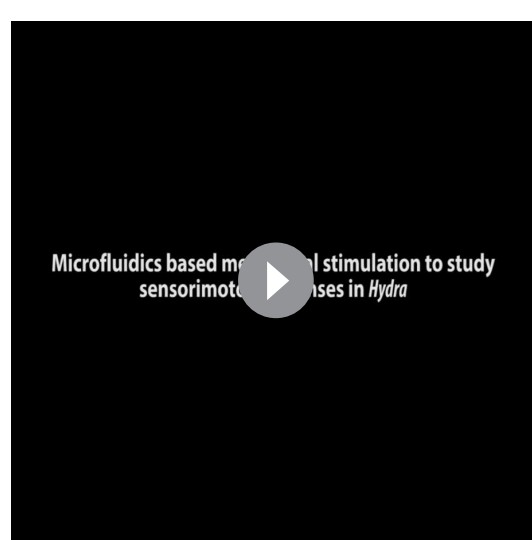

**Video 1.** Microfluidic system to study mechanosensory response in *Hydra*. Dashed blue circle indicates the microfluidic valve that presses down on *Hydra*. Mechanosensory response from neurons and epitheliomuscular cells is shown.
https://elifesciences.org/articles/64108#video1

calcium signals generated by a small number of neurons in the hypostome and body column and a striking co-activation of many neurons in the ectodermal peduncle nerve ring (*Figure 2b*, *Videos 2* and *3*). This nerve ring activity appeared as either a single bright calcium spike (or 'contraction pulse') or a volley of bright calcium spikes (or 'contraction burst'). We also found that the calcium-sensitive fluorescence averaged over a region of interest (ROI) surrounding the peduncle faithfully represented the contraction pulses and bursts measured from individual neurons (*Figure 2—figure supplement 2*). When we analyzed single-neuron calcium dynamics from this peduncle nerve ring, we found extremely high correlated activity as previously reported for contractions pulses and bursts (*Figure 2—figure supplement 2*; *Dupre and Yuste, 2017*; *Badhiwala et al., 2018*). Given the similarity of these data between large and small ROIs, we chose to use the peduncle ROI to measure neuronal contraction bursts and pulses because it does not require single-neuron tracking, which significantly increased the throughput of our data analysis. We further confirmed that this signal is not the result of motion artifacts by measuring fluorescence from *Hydra* (nGreen) that express GFP pan-neuronally using a similar ROI. In that case, we did not see the strong fluorescence signals associated with contraction pulses and bursts (*Figure 2—figure supplement 3e, f*). Body length proved to be an unreliable quantification of contractions due to the stimulation artifacts (*Figure 2—figure supplement 3d*); however, we were able to accurately measure muscle activity associated with contractions by imaging calcium spikes in the epithelial muscle cells (*Figure 2—figure supplements 4* and *5* and *Videos 4* and *5*). Based on these experiments, we define *Hydra's* 'mechanosensory response' as calcium spikes in neural activity from the peduncle ROI and the associated calcium spikes in the epithelial muscles from the whole body if they occur within 1 s of mechanical stimulation onset (*Figure 2c, d* and *Figure 2—figure supplement 6*).

Using the neuronal fluorescence calcium imaging described above, we found that the probability of the mechanosensory response depends on the intensity of the stimulus, which is consistent with many psychometric functions (*Figure 2e*). *Hydra* were five times more likely to contract within 1 s of receiving a strong mechanical stimulus than during a random 1 s interval without a stimulus (stimulus valve pressure 20 and 25 psi; response probability = 0.60 ± 0.06 and 0.77 ± 0.08, mean ± SEM, respectively; no stimulus valve pressure = 0 psi; response probability = 0.11 ± 0.01, mean ± SEM; *Figure 2e, h*). During mild stimuli, there was a slight increase (~2×) in response probability above the spontaneous activity, although this increase was not statistically significant compared to spontaneous contraction bursts or pulses (valve pressure 10 and 15 psi; response probability = 0.23 ± 0.01 and 0.24 ± 0.03, mean ± SEM, respectively; *Figure 2e, h*). We found that *Hydra* did not respond to a weak mechanical stimulus that corresponded to a valve pressure of 5 psi (response probability = 0.09 ± 0.02, mean ± SEM; *Figure 2e, h*).

Further analysis of the calcium activity pattern revealed that the single contraction pulses

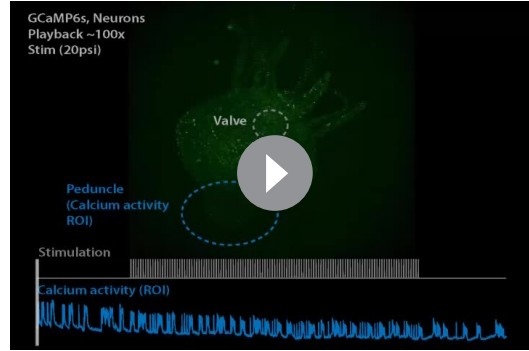

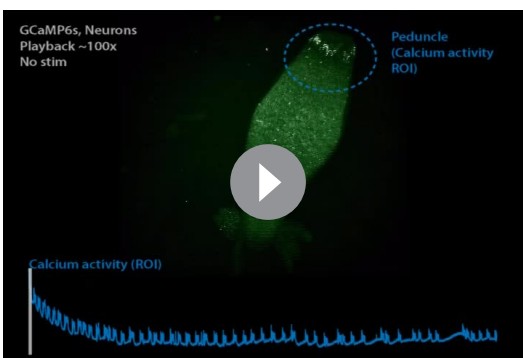

**Video 2.** Spontaneous neural calcium activity in normal animals. Dashed blue circle indicates the region of interest (ROI) used for calcium trace shown in blue (bottom) (playback 100×).
https://elifesciences.org/articles/64108#video2

**Video 3.** Stimulated neural calcium activity in normal animals. Dashed blue square indicates the region of interest (ROI) used for calcium trace shown in blue (bottom). Dashed white circle indicates the location of the valve that presses down on *Hydra* when inflated. Gray trace shows the stimulus protocol, where vertical lines indicate valve 'on' times. Stimulus applied beginning t = ~20 min and ends t = ~80 min. Valve is 'on' for 1 s and 'off' for 30 s (playback 100×).
https://elifesciences.org/articles/64108#video3

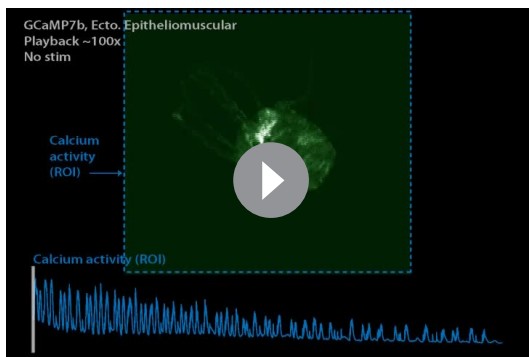

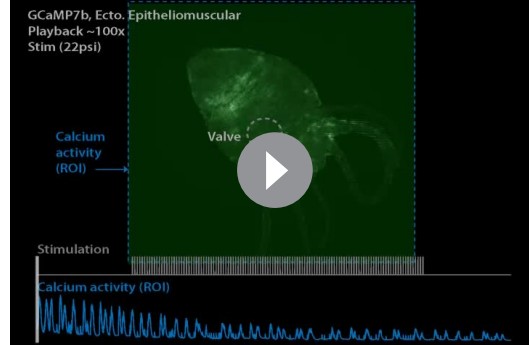

**Video 4.** Spontaneous endodermal epitheliomuscular calcium activity in normal animals. Dashed blue square indicates the region of interest (ROI) (entire frame) used for the calcium trace shown in blue (bottom) (playback 100×).
https://elifesciences.org/articles/64108#video4

**Video 5.** Stimulated endodermal epitheliomuscular calcium activity in normal animals. Dashed blue square indicates the region of interest (ROI) (entire frame) used for calcium trace shown in blue (bottom). Dashed white circle indicates the location of the valve that presses down on *Hydra* when inflated. Gray trace shows the stimulus protocol, where vertical lines indicate valve 'on' times. Stimulus applied beginning t = ~20 min and ends t = ~80 min. Valve is 'on' for 1 s and 'off' for 30 s (playback 100×).
https://elifesciences.org/articles/64108#video5

(calcium spikes in the peduncle neurons) were more frequent when we repeated mechanical stimulation every 31 s for 1 hr. While spontaneous contraction pulses or bursts were observed roughly once every minute in microfluidics, when we stimulated *Hydra* with a strong mechanical stimuli, we found that the frequency nearly matched the 31 s between stimuli (Interval between spontaneous contraction bursts or pulses = 72.98 s ± 4.58, mean ± SEM; 66.68 ± 5.73 s median ± SE; Interval between stimulated contraction bursts or pulses = 20 psi, 38.28 ± 1.77 s, mean ± SEM; 31.14 ± 2.21 s, median ± SE; 25 psi, 32.20 ± 1.04 s, mean ± SEM; 31.14 ± 1.30 s, median ± SE; *Figure 2f, h*). While the majority of spontaneous calcium spikes formed bursts, stimulated calcium spikes were roughly three times more likely to be a single contraction pulses (percentage of spontaneous spiking activity that is a single contraction pulse = 0 psi, 16.87 ± 5.38% mean ± SEM; percentage of stimulated spiking activity that is a single contraction pulse = 20 psi, 56.84 ± 2.48%; 25 psi, 62.50 ± 5.23%, mean ± SEM; *Figure 2g, h*).

## *Hydra* sensitivity to mechanical stimuli is lowest near the aboral end

Because of the diffuse neural architecture of *Hydra*, we expected each patch of *Hydra* tissue to be equally responsive to mechanical stimuli. However, when we stimulated transgenic *Hydra* expressing GCaMP6s (N = 3 whole animals, stimulated 40 times per region with 22 psi) at three different regions along their body (oral, middle body, and aboral, stimulated 40 times at each region with 22 psi), we found the aboral end of *Hydra* to be less sensitive than the center of the body (aboral region response probability = 0.1 ± 0.025; mid-body region response probability = 0.42 ± 0.025; mean ± SEM; p<0.01; *Figure 2—figure supplement 7a*). Epitheliomuscular calcium imaging (N = 8 whole animals expressing GCaMP7b in endodermal epitheliomuscular cells, stimulated 40 times in body column region with 22 psi) also showed that the sensitivity to mechanical stimulation generally decreases towards the aboral end of the *Hydra* (*Figure 2—figure supplement 7b*). Furthermore, we found that the difference in sensitivity along the body was not an artifact due to differently sized *Hydra* experiencing different pressures from the microfluidic valves. We observed no statistically significant trend between animal size and response probability (*Figure 2—figure supplement 7c*). These findings, combined with the transcriptional analysis and in situ hybridizations that indicate higher density of sensory neurons in the oral half of the *Hydra* (*Siebert et al., 2019*), suggest that the oral end may be more sensitive to mechanical stimuli. Unlike other organisms that have unique motor responses like reversals or acceleration depending on the location of mechanical stimuli (e.g., *C. elegans*) (*Chalfie et al., 1985*; *Wicks et al., 1996*), we observed that the same motor program was initiated regardless of where on the body the animal was touched. The only difference

we observed was the fact that the response probability depended on where along the oral and aboral axis we delivered the mechanical stimulus.

## *Hydra*'s mechanosensory response is mediated by electrically coupled cells

Based on the latency of the mechanosensory response, we hypothesized that sensorimotor information is transmitted by electrical activity in the *Hydra* and not by passive calcium diffusion through epithelial cells. This hypothesis is supported by the fact that aneural *Hydra,* which no longer spontaneously contract in the absence of stimuli, are capable of aversive contractile response to touch; however, their responses are slow and require strong mechanical stimulus (*Campbell et al., 1976*; *Takaku et al., 2014*). To further test our hypothesis, we created a transgenic *Hydra* strain that expresses the calcium indicator GCaMP7b (under the EF1α promoter, see Materials and methods) in the endodermal epitheliomuscular cells. During body contractions, both endodermal and ectodermal epitheliomuscular cells are co-activated (*Wang et al., 2020*). With this transgenic line we measured contraction pulses and contraction bursts by averaging calcium activity in all epitheliomuscular cells (*Figure 2—figure supplements 3* and *5*). We hypothesized that if the mechanosensory response was primarily mediated by calcium diffusion through epitheliomuscular cells, we would expect to see propagation of calcium activity from the site of the stimulation. However, this was not the case; we observed fast propagation of calcium activity throughout the entire endoderm. Imaging calcium activity in the peduncle neurons showed that changes in neural activity correlated with the observed changes in epithelial muscle cells with increasing stimulus intensity (*Figure 2* and *Figure 2—figure supplement 5*). This suggests that the mechanosensory response is indeed mediated by the electrically coupled cells. Although *Hydra*'s behavioral responses are slow compared to other invertebrates, the 0.5 s average response time in our data could not be explained by calcium diffusion alone, which would take ~100 s to travel the average distance of ~0.5–1 mm between the stimulation site and the peduncle or hypostome (*Figure 2—figure supplement 8a*). Our data show that the calcium signals from the peduncle neurons or endodermal muscles start increasing within ~0.1–0.2 s following stimulation onset, reaching peak fluorescence at ~0.5–0.6 s regardless of the stimulus intensity (*Figure 2—figure supplement 8b–e*). We found no difference in the calcium response times between neural and muscle calcium imaging, though this could be influenced by the dynamics of the calcium indicator, which typically cannot give information about latencies less than 50 ms (*Chen et al., 2013*).

## Aboral neurons are not necessary for mechanosensory response in *Hydra*

Having quantitatively established *Hydra's* sensorimotor response to mechanical stimuli, we next asked if specific regions of the nervous system play primary roles in mediating this response. Patch-clamp electrophysiology of individual neurons in *Hydra* has thus far been unsuccessful despite attempts by many research groups, and minimally invasive, cell-type specific neuromodulation techniques such as optogenetics have yet to be developed for *Hydra*. However, the animal's regenerative abilities allow us to resect large portions of tissue without killing the animal, thus we can borrow from the tradition of lesioning brain regions to study their functions (*Rushforth et al., 1963*; *Vaidya et al., 2019*; *Krug et al., 2015*; *Passano and McCullough, 1964*; *Pierobon, 2015*).

Because the *Hydra* body plan is radially symmetric with cell types primarily varying along the oral-aboral axis of the body column (*Figure 1*), we chose to make axial cuts across the body column to remove select neuronal populations from the animal. Our rationale was that these resections would remove entire or nearly entire groups of neuronal cell types. We then allowed ~6–12 hr for the animal to recover. This recovery time helps to reduce the confounding contributions from initial tissue regeneration and allows animals to recover enough to tolerate microfluidic immobilization. This period is long enough to allow the wounds to close, the molecular response to injury to be completed (*Cazet and Juliano, 2020*; *Tursch et al., 2020*), and the initial molecular events of regeneration to start; but it is *not* long enough for the animal to regenerate lost neurons, which takes approximately 30–72 hr (*Figure 3—figure supplement 1*; *Itayama and Sawada, 1995*; *Pierobon, 2015*; *Bode, 2003*; *Goel et al., 2019*). We confirmed that 6–12 hr after resection the animals indeed showed loss of specific neuronal cell types by measuring the expression levels of subtype-

specific neuronal markers via qPCR (*Figure 3—figure supplement 2*). To limit the stress from microfluidic immobilization that could exacerbate the resection wounds and affect activity, we shortened the duration of these experiments for the majority of the animals (40 min total, 20 min of no stimulation, 20 min of stimulation – valve on 1 s at 22 psi or 0 psi, off 30 s). Only three animals per each condition (stimulated and non-stimulated, and five different resections) were experimented on with a longer duration protocol as used previously (100 min total, 20 min no stimulation, 60 min of stimulation – valve on 1 s at 22 psi or 0 psi, off 30 s, 20 min no stimulation; *Figure 3—figure supplement 3*).

We began resection studies by removing the peduncle and basal disk to create a 'footless' *Hydra.* We hypothesized that aboral neurons may be important for coordinating and enhancing body contractions (*Figure 3*). We based this hypothesis on the fact that aboral neuron activity has a strong correlation with body contractions (*Dupre and Yuste, 2017*; *Badhiwala et al., 2018*). In addition, the neuropeptide Hym-176C has been shown to induce ectodermal muscle contractions and is selectively expressed in the ectodermal peduncle neurons (*Siebert et al., 2019*; *Yum et al., 1998*; *Noro et al., 2019*; *Klimovich et al., 2020*). Finally, the presence of gap junction protein innexin-2 in aboral neurons could facilitate fast electrical conductions that allows these neurons to fire synchronously (*Siebert et al., 2019*; *Takaku et al., 2014*). This could be necessary for enhancing neuromuscular signaling for body contractions. Because 'footless' *Hydra* lacked peduncle neurons that we had used previously to measure contraction pulses and bursts (*Figure 3—figure supplement 3d–f* and *Videos 6–9*), we performed these experiments using a transgenic *Hydra* line expressing GCaMP7b in the endodermal epitheliomuscular cells, which allowed us to measure contraction pulses and bursts by averaging calcium activity in all the epitheliomuscular cells (using whole-frame ROI, which is more robust to motion artifacts than peduncle ROI; *Figure 2—figure supplement 3*, *Figure 3a, b*, *Figure 3—figure supplement 3*).

Because neurons in the foot fire synchronously with body contractions, we expected 'footless' animals to show significant changes in contraction behaviors (calcium spiking activity) to be significantly affected by their removal (*Dupre and Yuste, 2017*; *Badhiwala et al., 2018*; *Shimizu and Fujisawa, 2003*), but this was not what we observed. Surprisingly, our experiments with 'footless' animals showed that the aboral nerve ring was not required to regulate spontaneous contraction bursts or pulses or mechanosensory responses. After we removed the peduncle network in *Hydra*, we found that the increase in contraction burst or pulse activity with stimuli (or mechanosensory response) in 'footless' individuals was similar to the increase in activity observed in whole individuals ('footless' N = 8 animals stimulated 20 min, Cohen's d = 1.89, Cliff's delta = 0.81, p<0.01; whole N = 8 animals stimulated 20 min, Cohen's d = 1.80, Cliff's delta = 1.00, p<0.01; *Figure 3b, c*). Furthermore, we found no significant difference in either the spontaneous contraction probability or the mechanosensory response probability of 'footless' animals and whole animals ('footless' N = 3 animals not stimulated, spontaneous contraction probability = 0.13 ± 0.01, mean ± SEM; 'footless' N = 3 animals stimulated for 60 min, mechanosensory response probability = 0.88 ± 0.05, mean ± SEM; whole N = 3 animals not stimulated, spontaneous contraction probability = 0.15 ± 0.01, mean ± SEM; whole N = 3 animals stimulated for 60 min, mechanosensory response probability = 0.73 ± 0.07, mean ± SEM; *Figure 3—figure supplement 3b, c*, *Videos 4*, *5*, *10,* and *11*).

## Oral neurons play a major role in mechanosensory response in *Hydra*

While the mechanosensory response in *Hydra* remained unaffected with the removal of the aboral nerve ring, we found that removal of the hypostome and tentacles (or 'headless' *Hydra*) resulted in significant changes in both the mechanosensory response and spontaneous contraction bursts or pulses. When we measured the mechanosensory response in 'headless' *Hydra,* we found that the animals still responded to mechanical stimulation with a significant increase in their contraction bursts or pulses; however, they did so with a lower magnitude compared to whole and 'footless' individuals ('headless' N = 8 animals stimulated for 20 min, p<0.01, Cohen's d = 1.37, Cliff's delta = 1.00; *Figure 3b, c*). Specifically, the 'headless' *Hydra* responded with (>2×) lower probability compared to whole animals, and they also showed a (>3×) lower probability of spontaneous contraction bursts and pulses ('headless' N = 3 animals stimulated for 60 min, mechanosensory response probability = 0.29 ± 0.02, mean ± SEM; 'headless' N = 3 animals not stimulated, spontaneous contraction probability = 0.05 ± 0.01, mean ± SEM; *Figure 3—figure supplement 3b, c*, *Videos 12* and *13*).

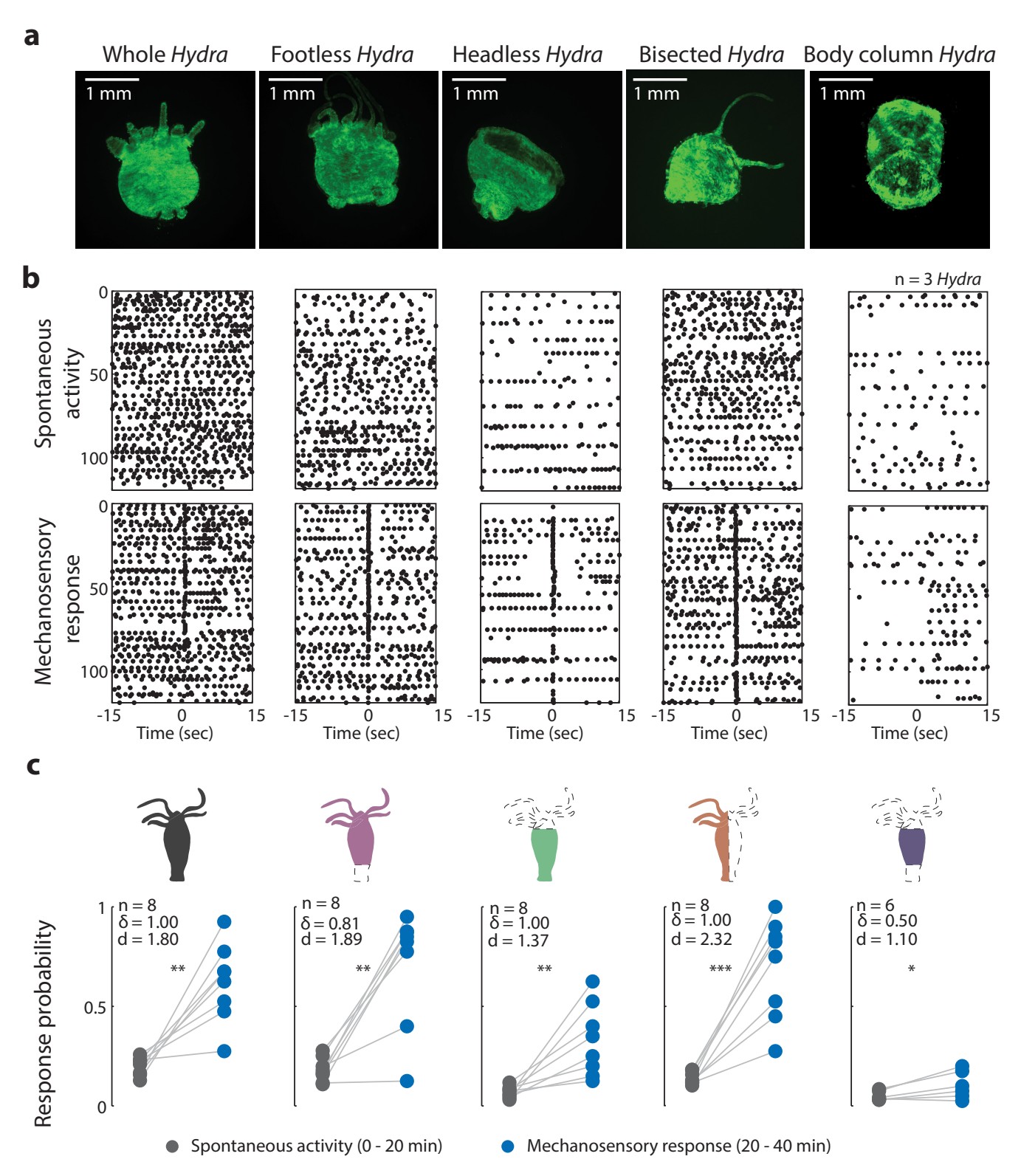

**Figure 3.** Oral region is important for mechanosensory response. (a) Representative images of resection preparations (6–12 hr post-resections) of transgenic *Hydra* during contraction (GCaMP7b, endodermal epitheliomuscular cells): whole (or control), footless, headless, bisected, and body column animal. Entire frame region of interest (ROI) was used for analysis of the whole-body epithelial calcium activity. (b) Representative raster plot of stimulus time-aligned calcium spikes from three animals with multiple trials superimposed. Each black dot is a peak in calcium fluorescence identified as a
*Figure 3 continued on next page*

*Figure 3 continued*

contraction pulse. (c) Response probability, fraction of trials that have at least one calcium spike (also contraction pulse) occurring within 1 s of stimulation onset. Gray dots are the mean contraction probability during no stimulation (t = 0–20 min) calculated from 1 s window shifted by ~0.3 s over 30 s intervals. Blue dots are contraction probability during stimulation (t = 20–40 min) calculated from 1 s response window during valve 'on.' Light gray lines connect the probabilities for spontaneous contraction and mechanosensory response for each individual. Cartoon schematics of *Hydra* indicate the resections performed. Excised body regions are outlined with a dashed line and unfilled area. Color-filled body regions indicate the portion of *Hydra* retained for the experiment. p-values from a paired t-test indicated as follows: n.s. = not significant; *p<0.05; **p<0.01; ***p<0.001. Source data for the mechanosensory response in resected animals are available in *Figure 3—source data 1*.

The online version of this article includes the following source data and figure supplement(s) for figure 3:

**Source data 1.** Source file of mechanosensory response in resected animals.
**Figure supplement 1.** Regeneration of the peduncle network.
**Figure supplement 2.** RT-qPCR analyses of neuron subtype-specific gene expression in resected animals demonstrates loss of specific neuron subtypes.
**Figure supplement 3.** Mechanosensory response from endodermal epitheliomuscular cells and neurons in resected *Hydra*.
**Figure supplement 4.** Contraction activity in non-stimulated animals.
**Figure supplement 5.** Hypostome and peduncle nerve rings work together to coordinate contractile behavior.

To verify the reduction in mechanical response probability was specific to removing the oral network of neurons, and not simply the result of injury, we performed experiments with animals that we cut longitudinally along the body axis ('bisected' *Hydra*) to remove approximately the same amount of tissue while preserving the neuronal subtypes in both the oral and aboral networks. We found that these longitudinally 'bisected' animals showed mechanosensory responses that were not significantly different from that of the whole, 'headless,' or 'footless' animals ('bisected' N = 3 animals stimulated for 60 min, mechanosensory response probability = 0.54 ± 0.14, mean ± SEM; *Figure 3— figure supplement 3b, c*). However, the magnitude of increase in contraction bursts or pulses activity due to stimulation in 'bisected' individuals, though lower than whole and 'footless' individuals, was larger than 'headless' individuals ('bisected' N = 8 animals stimulated for 20 min, p<0.001, Cohen's d = 2.32, Cliff's delta = 1.0). This suggests that our observations in 'headless' *Hydra* indeed depended upon the types of neurons removed during the headless resection and not simply an injury response. We also found that these longitudinally 'bisected' animals had a lower probability of spontaneous contraction bursts or pulses than the whole animals but higher than 'headless' animals. This suggests that the entire network needs to be intact for normal contraction bursts or pulses activity, and the loss of roughly half the network leads to some reduction in contractile activity ('bisected' N = 3 animals not stimulated, spontaneous contraction probability = 0.08 ± 0.02, mean ± SEM; *Figure 3—figure supplement 3b, c*, *Videos 14* and *15*).

We next asked if the body column alone is sufficient to mediate the mechanosensory response. To answer this question, we completely removed both oral and aboral regions. In 'body column' animals, we found significant reduction in both the mechanosensory response and spontaneous contraction bursts and pulses relative to whole animals. The 'body column' animals had a mechanosensory response probability that was not different from the mechanosensory response in 'headless' animals, while the spontaneous contraction bursts and pulses probability was lower compared to that of 'headless' animals ('body column' N = 3 animals stimulated for 60 min, mechanosensory response probability = 0.19 ± 0.08; 'body column' N = 3 animals not stimulated, spontaneous contraction probability = 0.02 ± 0.01 mean ± SEM; *Figure 3—figure supplement 3b, c*). Although we found a significant increase in contraction bursts and pulses with stimulation as compared to

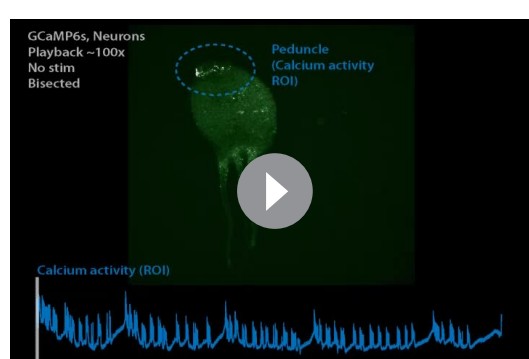

**Video 6.** Spontaneous neural calcium activity in longitudinally bisected animals. Dashed blue circle indicates the region of interest (ROI) used for calcium trace shown in blue (bottom) (playback 100×).
https://elifesciences.org/articles/64108#video6

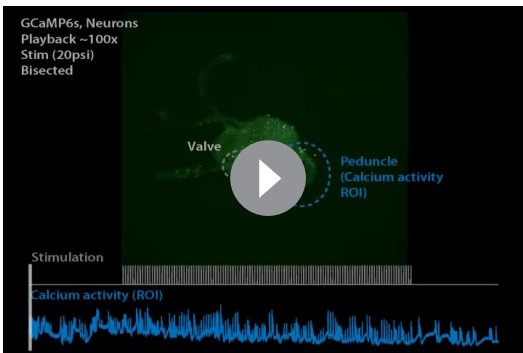

**Video 7.** Stimulated neural calcium activity in longitudinally bisected animals. Dashed blue circle indicates the region of interest (ROI) used for calcium trace shown in blue (bottom). Dashed white circle indicates the location of the valve that presses down on *Hydra* when inflated. Gray trace shows the stimulus protocol, where vertical lines indicate valve 'on' times. Stimulus applied beginning t = ~20 min and ends t = ~80 min. Valve is 'on' for 1 s and 'off' for 30 s (playback 100×).
https://elifesciences.org/articles/64108#video7

**Video 8.** Spontaneous neural calcium activity in headless animals. Dashed blue circle indicates the region of interest (ROI) used for calcium trace shown in blue (bottom). Animal is less active without the head and fewer spikes in calcium activity of peduncle neurons are observed (playback 100×).
https://elifesciences.org/articles/64108#video8

spontaneous contraction bursts and pulses activity in 'body column' individuals similar to all resections, the magnitude of the increase was the lowest in 'body column' animals (even lower than 'headless' animals) ('body column' N = 6 animals stimulated for 20 min, p<0.05, Cohen's d = 1.10, Cliff's delta = 0.50; *Figure 3b, c*, *Videos 16* and *17*). Moreover, we did not observe significant increases in contraction bursts and pulses over the same time period (comparing activity from 0 to 20 min with activity from 20 to 40 min, see Materials and methods) in non-stimulated animals, suggesting that the higher probability of contraction bursts and pulses was in fact due to mechanical stimulation (*Figure 3—figure supplement 4*). Thus, based on the comparison between the probability of spontaneous contraction bursts and pulses and mechanosensory response in all resections, we found that the 'body column' animals had a weak response to touch despite their slightly increased contraction bursts and pulses probability with mechanical stimulation.

## Oral and aboral networks show different patterns of activity during spontaneous contractions compared to mechanically stimulated contractions

To identify how the activity of neurons in the oral and aboral networks coordinate spontaneous and stimulated responses, we manually tracked the calcium activity of several neurons in the oral and aboral regions (20 min no stimulation, 10 min stimulation 1 s every 31 s, n = 3 *Hydra*; *Figure 4a, b*, *Figure 4—figure supplements 1* and *2*). We found that there were at least two independent networks of neurons based on a

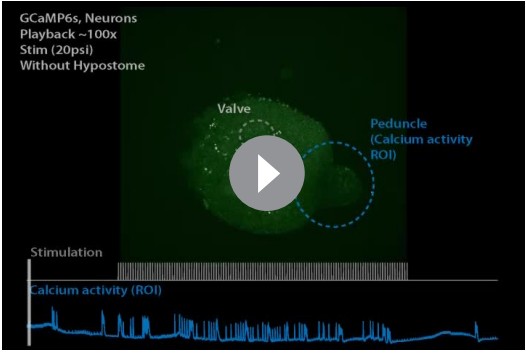

**Video 9.** Stimulated neural calcium activity in headless animals. Dashed blue circle indicates the region of interest (ROI) used for calcium trace shown in blue (bottom). Dashed white circle indicates the location of the valve that presses down on *Hydra* when inflated. Gray trace shows the stimulus protocol, where vertical lines indicate valve 'on' times. Stimulus applied beginning t = ~20 min and ends t = ~80 min. Valve is 'on' for 1 s and 'off' for 30 s. Animal is less active without the head and fewer spikes in calcium activity of peduncle neurons are observed without stimulation. After stimulation begins, more single-pulse contractions occur (playback 100×).
https://elifesciences.org/articles/64108#video9

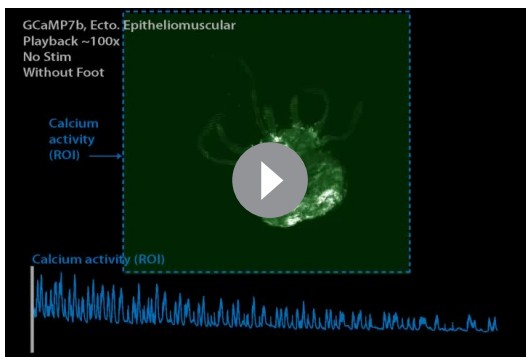

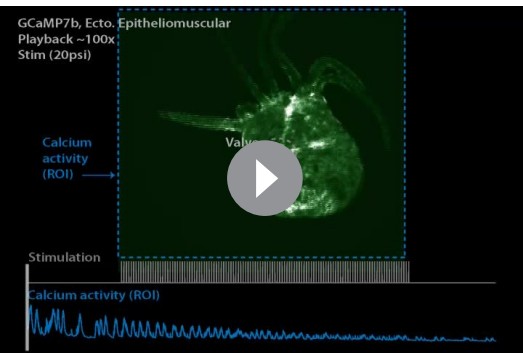

**Video 10.** Spontaneous endodermal epitheliomuscular calcium activity in footless animals. Dashed blue square indicates the region of interest (ROI) (entire frame) used for the calcium trace shown in blue (bottom) (playback 100×).
https://elifesciences.org/articles/64108#video10

**Video 11.** Stimulated endodermal epitheliomuscular calcium activity in footless animals. Dashed blue square indicates the region of interest (ROI) (entire frame) used for calcium trace shown in blue (bottom). Dashed white circle indicates the location of the valve that presses down on *Hydra* when inflated. Gray trace shows the stimulus protocol, where vertical lines indicate valve 'on' times. Stimulus applied beginning t = ~20 min and ends t = ~80 min. Valve is 'on' for 1 s and 'off' for 30 s (playback 100×).
https://elifesciences.org/articles/64108#video11

correlation analysis (*Figure 4c*). Specifically, we time-aligned the calcium activity with either spontaneous contractions or mechanical stimulation events to classify these groups of neurons based on their activity (*Figure 4d, e, g*). One group of correlated neurons found throughout the entire body showed averaged calcium activity less than 1 s after a mechanical stimulus and spontaneous activity that is consistent with previously reported contraction burst (CB) neurons (*Dupre and Yuste, 2017*). We plot calcium dynamics of these CB neurons as shades of blue in *Figure 4* and *Figure 4—figure supplements 1* and *2*. These neurons show bursts of activity that are synchronized with muscle contractions and show calcium activity that is highly correlated with the average peduncle ROI. In addition to these CB neurons, we found other groups of correlated neurons with average calcium activity that is independent of the CB network. One group showed a distinctive pattern of activity following mechanical stimulation, but no distinctive activity associated with spontaneous contractions. Specifically, this group of neurons near the oral end responded approximately 10 s after mechanical stimulation (*Figure 4e, g*). We found these putative 'mechanically responsive (MR) neurons' in all three *Hydra* we analyzed and plot their calcium dynamics as shades of red in *Figure 4* and *Figure 4—figure supplements 1* and *2*. The fact that these MR neurons do not show activity associated with spontaneous contractions clearly indicates that they are not a part of the CB network, but rather these two distinct networks (CB and MR) are involved in the *Hydra's* response to mechanical stimulation. We also note that these MR neurons do not fire periodically as would be expected for the rhythmic potential (RP) network (*Dupre and Yuste, 2017*). In addition to CB and MR neurons, in *Hydra* 2 (*Figure 4—figure supplement 1*) and 3 (*Figure 4—figure supplement 2*) we also found neurons that were not associated with either the MR network or the CB network. These neurons we labeled as 'unspecified

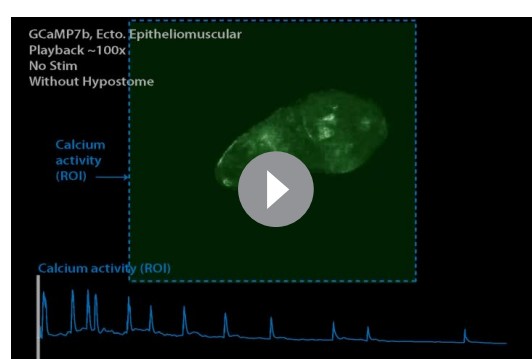

**Video 12.** Spontaneous endodermal epitheliomuscular calcium activity in headless animals. Dashed blue square indicates the region of interest (ROI) (entire frame) used for the calcium trace shown in blue (bottom). Animal is less active without the head and fewer spikes in calcium activity of the whole body endodermal epitheliomuscular cells are observed (playback 100×).
https://elifesciences.org/articles/64108#video12

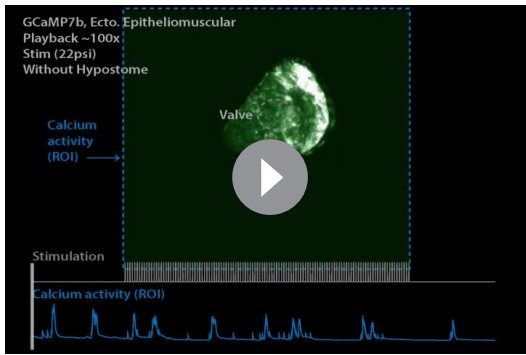

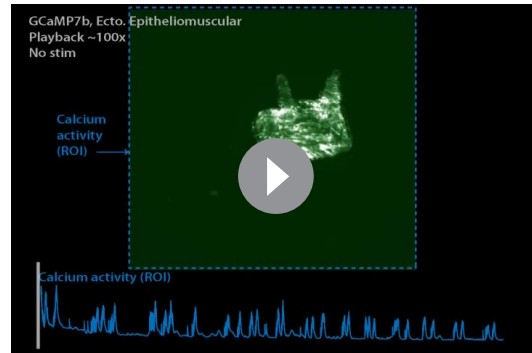

**Video 13.** Stimulated endodermal epitheliomuscular calcium activity in headless animals. Dashed blue square indicates the region of interest (ROI) (entire frame) used for calcium trace shown in blue (bottom). Dashed white circle indicates the location of the valve that presses down on *Hydra* when inflated. Gray trace shows the stimulus protocol, where vertical lines indicate valve 'on' times. Stimulus applied beginning t = ~20 min and ends t = ~80 min. Valve is 'on' for 1 s and 'off' for 30 s. Animal is less active without the head and fewer spikes in calcium activity of epitheliomuscular cells are observed without stimulation. After stimulation begins, more single-pulse contractions occur (playback 100×).
https://elifesciences.org/articles/64108#video13

**Video 14.** Spontaneous endodermal epitheliomuscular calcium activity in bisected animals. Dashed blue square indicates the region of interest (ROI) (entire frame) used for the calcium trace shown in blue (bottom). Animal is less active without the head and fewer spikes in calcium activity of the whole body endodermal epitheliomuscular cells are observed (playback 100×).
https://elifesciences.org/articles/64108#video14

groups' do not appear to be a part of the CB or RP networks previously characterized nor the MR neurons we identify here.

These data suggest that there are at least two separate pathways involved in the mechanosensory response. The first involves the CB neu-

rons and muscle contractions. The second network involving the MR neurons responds more slowly, with a latency of several seconds. Because the activity of the MR neurons occurs after the contraction, their role in the behavioral response remains unclear.

## Discussion

Our experiments with 'footless,' 'headless,' 'bisected,' and 'body column' animals show that mechanosensory and spontaneous behaviors are regulated by neural ensembles that are localized to select regions of the animal; however, some properties of the mechanosensory response may be evenly distributed throughout the body. We demonstrate that localized touch produces an increased calcium activity in both peduncle neurons (*Figure 2*) and endodermal epitheliomuscular cells (*Figure 3*), which is associated with body contractions (*Figure 2—figure supplement 4*; *Badhiwala et al., 2018*). We identified at least two neuronal networks (MR and CB network) with distinct neuronal activities mediating the stimulated responses (*Figure 4*), where the CB network of neurons show fast calcium responses, and the MR neurons show slower calcium response. It is possible that the MR neurons,

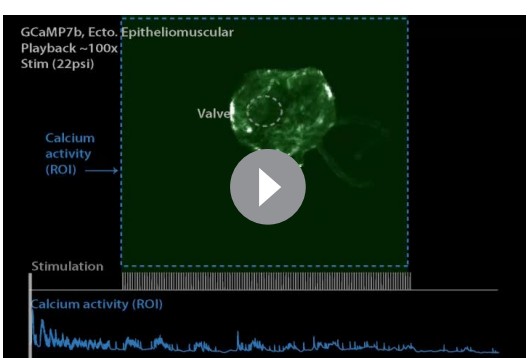

**Video 15.** Stimulated endodermal epitheliomuscular calcium activity in bisected animals. Dashed blue square indicates the region of interest (ROI) (entire frame) used for calcium trace shown in blue (bottom). Dashed white circle indicates the location of the valve that presses down on *Hydra* when inflated. Gray trace shows the stimulus protocol, where vertical lines indicate valve 'on' times. Stimulus applied beginning t = ~20 min and ends t = ~80 min. Valve is 'on' for 1 s and 'off' for 30 s. Animal is less active without the head and fewer spikes in calcium activity of epitheliomuscular cells are observed without stimulation. After stimulation begins, more single-pulse contractions occur (playback 100×).
https://elifesciences.org/articles/64108#video15

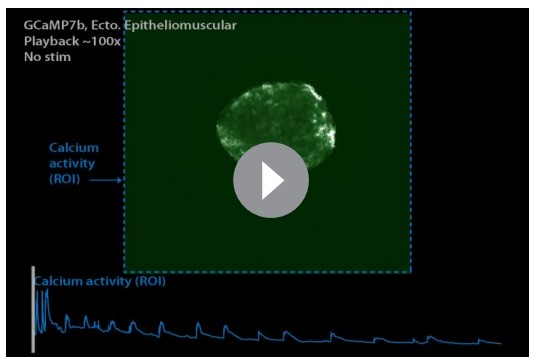

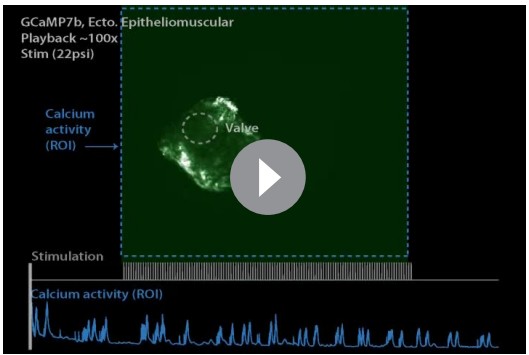

**Video 16.** Spontaneous endodermal epitheliomuscular calcium activity in body column animals. Dashed blue square indicates the region of interest (ROI) (entire frame) used for the calcium trace shown in blue (bottom). Animal is less active without the head and fewer spikes in calcium activity of the whole body endodermal epitheliomuscular cells are observed (playback 100×).
https://elifesciences.org/articles/64108#video16

**Video 17.** Stimulated endodermal epitheliomuscular calcium activity in body column animals. Dashed blue square indicates the region of interest (ROI) (entire frame) used for calcium trace shown in blue (bottom). Dashed white circle indicates the location of the valve that presses down on *Hydra* when inflated. Gray trace shows the stimulus protocol, where vertical lines indicate valve 'on' times. Stimulus applied beginning t = ~20 min and ends t = ~80 min. Valve is 'on' for 1 s and 'off' for 30 s. Animal is less active without the head and fewer spikes in calcium activity of epitheliomuscular cells are observed without stimulation. After stimulation begins, more single-pulse contractions occur (playback 100×).
https://elifesciences.org/articles/64108#video17

primarily found in the oral end (N = 3 *Hydra*), may consist of ec4 neurons, which are the only neuronal subtype in the oral end with an unknown function. The other orally located ecto-dermal neuron populations, ec1B and ec3C, are suggested to belong to the CB and RP1 circuits, respectively (*Siebert et al., 2019*). *Hydra's* responsiveness (i.e., probability of response to a mechanical stimuli) depends on the stimulus intensity (*Figure 2*). Interestingly, we found that the 'headless' and 'footless' animals can still respond to mechanical stimuli despite missing an entire regional neuronal network; however, 'headless' animals show reduced responsiveness (*Figure 3* and *Figure 3—figure supplement 3*). The 'body column' animals missing both regional neuronal networks have the most dramatically reduced responsiveness, suggesting that these two neuronal networks work together and play compensatory roles in mediating the mechanosensory response.

Surprisingly, although the activity of the peduncle nerve ring is strongly associated with spontaneous contractions, these neurons are not necessary for body contraction and response to mechanical stimulation. This raises the question of what role the peduncle nerve ring plays in *Hydra* behavior. One possible explanation is that this nerve ring coordinates body contractions by enhancing the neural signal to epithelial cells. An additional 4 s reduction in body contraction duration in 'body column' animals compared to headless animals supports the idea that the peduncle nerve ring is also involved in coordination of contractile behavior (*Figure 3—figure supplement 5*). Furthermore, calcium activity propagates from the foot to the hypostome in whole animals during body contractions (*Szymanski and Yuste, 2019*), supporting a hypothesis that the peduncle network of neurons may be motor neurons.

Based on these observations, we propose a simple model for sensorimotor information flow in *Hydra* where we consider the hypostome as an integration point where sensory and motor information converge. The information is then communicated to the peduncle where it is amplified for coordinated whole-body control. There may be sufficient functional redundancy between the hypostome and peduncle regions such that removal or damage to one of them is well tolerated in *Hydra*. Moreover, the diffuse network in the body column may retain minimal processing capabilities needed for weak sensorimotor responses. The fast and slow calcium responses from the CB and MR neurons, respectively, indeed support the hypothesis that *Hydra* have multiple, separate pathways for behavioral response.

Overall, the quantitative characterization of *Hydra's* sensorimotor responses reported here helps to build the foundation for a more comprehensive investigation of information processing in *Hydra* –

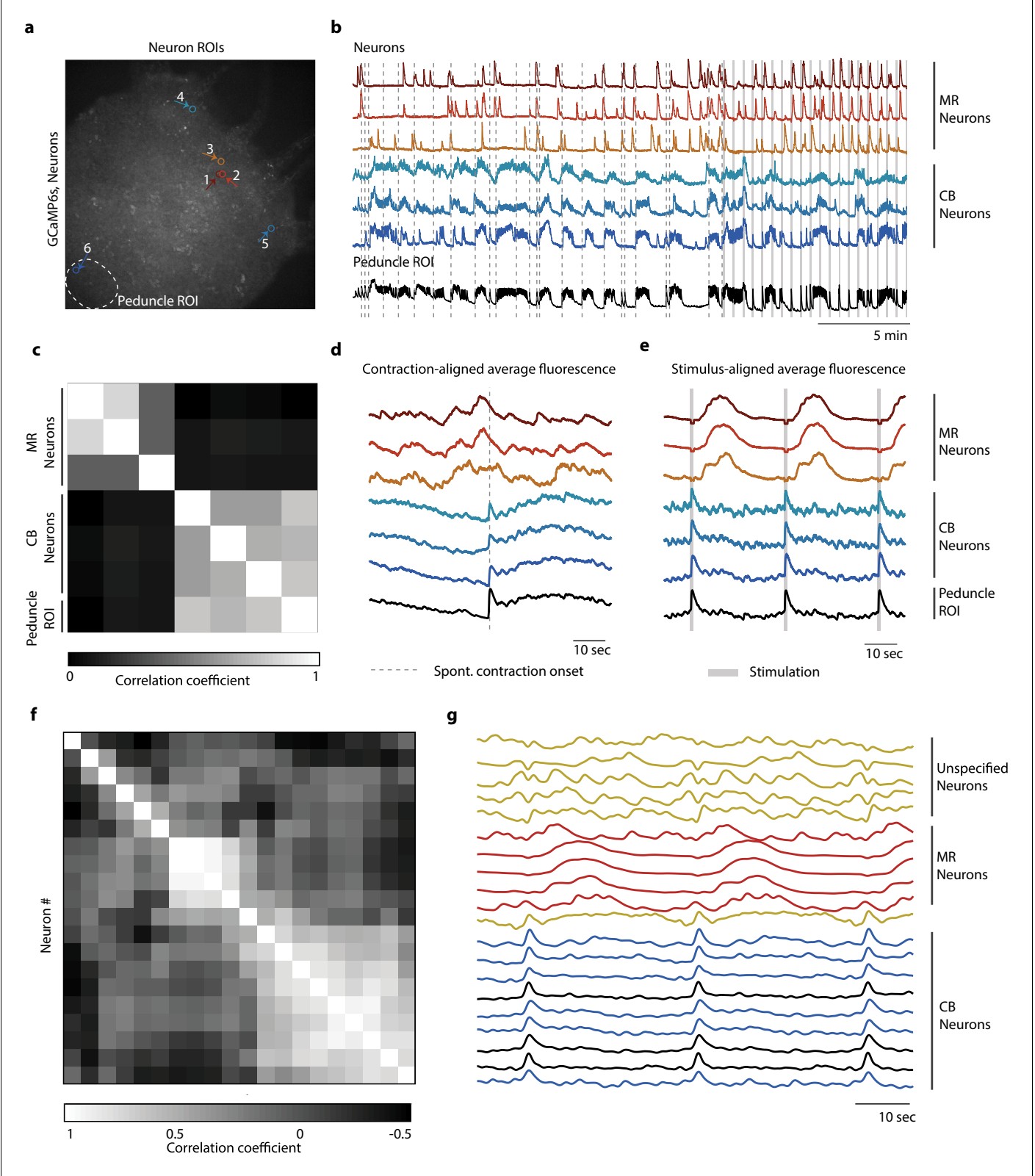

**Figure 4.** Distinct networks of neurons involved in spontaneous and stimulated behaviors. (**a**) Fluorescent image of transgenic *Hydra* expressing GCaMP6s pan-neuronally. Individually tracked neurons regions of interest (ROIs) are indicated by arrows, and peduncle ROI is outlined with a white dashed circle. (**b**) Calcium fluorescence traces from single neurons (top six traces) and average calcium fluorescence from peduncle ROI (bottom trace). Mechanically responsive (MR) neurons are shown in shades of red. Contraction burst (CB) neurons are shown in shades of blue. (**c**) Heat map shows the

*Figure 4 continued on next page*

*Figure 4 continued*

correlation coefficients of individually tracked neurons and peduncle ROI from *Hydra* 1, with color bar at the bottom. Correlation was computed using the entire 30 min calcium fluorescence from single neurons shown in (**b**). (**d**) Average calcium fluorescence traces from each of the neurons and peduncle ROI during spontaneous behaviors time-aligned with the onset of spontaneous body contractions. Dashed line indicates the onset of body contraction. (**e**) Average calcium fluorescence traces from each of the neurons and peduncle ROI during stimulated behaviors time-aligned with the onset of mechanical stimulation. Gray-shaded rectangle indicates mechanical stimulation. (**f**) Heat map shows the correlation coefficients of (**g**) the average calcium fluorescence traces from each of the neurons and peduncle ROI (traces in black) from three different *Hydra* during stimulated behaviors time-aligned with the onset of mechanical stimulation. Three groups of neurons were identified across the three *Hydra*. CB neurons (traces in blue), MR neurons (traces in red), and unspecified neurons (traces in yellow). (**a–e**) Calcium fluorescence traces and correlation analysis (entire 30 min of activity, 20 min of spontaneous activity, and 10 min of stimulated activity) from one representative *Hydra*. (**f, g**) Correlation analysis of the average stimulus-aligned calcium fluorescence (30 s stimulation interval average from 10 min of stimulated activity) of each neuron pooled multiple *Hydra* (N = 3 *Hydra*) to highlight the three groups of neurons identified: CB, MR, and unspecified. Source data for fluorescent calcium activity from single neurons are available in *Figure 4—source data 1*.

The online version of this article includes the following source data and figure supplement(s) for figure 4:

**Source data 1.** source file for fluorescent calcium activity from single-neuron regions of interest (ROIs).
**Figure supplement 1.** Single-cell correlation analysis.
**Figure supplement 2.** Single-cell correlation analysis.
**Figure supplement 3.** Random shuffling of single-cell correlation analysis.

an animal with clear advantages to supplement commonly studied model organisms in neuroscience. Important next steps include developing mechanistic models to describe sensory information processing that supports our results. Although our experiments describe general information flow in *Hydra*, having a cellular-level control of neuronal activity would be a clear advantage for revealing the function of neuronal cell types as well as their functional connectivity in the sensorimotor circuits. We expect additional future work with optogenetic manipulation of specific neuronal subtypes combined with fast, volumetric and ratiometric imaging techniques will provide a more comprehensive approach for characterizing this sensorimotor processing in *Hydra*. Recognizing that calcium fluorescence imaging is limited in its ability to measure single spikes (*Huang et al., 2021*), we also expect other activity sensors (such as voltage indicators) will reveal what role the MR neurons play in behavioral response. Building upon the work reported here, one can then interrogate the roles of these regional networks with multiple sensory modalities, such as light and heat, to answer questions about how diffuse nervous systems may be capable of centralized information processing and multi-sensory integration. These studies may help reveal a comprehensive model for how internal states and external stimuli shape the behavioral repertoire in an organism with a highly dynamic neural architecture.

## Materials and methods

### *Hydra* strains and maintenance

*Hydra* were raised in *Hydra* medium at 18°C in a light-cycled (12 hr:12 hr; light:dark) incubator and fed with an excess of freshly hatched *Artemia nauplii* (Brine Shrimp Direct, Ogden, UT, #BSEP8Z) three times a week (protocol adapted from Steele lab). All experiments were performed at room temperature with animals starved for 2 days.

The transgenic line nGreen, kindly provided by Rob Steele, was generated by microinjecting embryos with a plasmid containing the *Hydra* actin promoter driving GFP expression (*Siebert et al., 2019*). The transgenic strains expressing GCaMP6s under the actin promoter in neurons and in ectodermal epitheliomuscles (Addgene plasmid: #102558) were developed by microinjections of the embryos by Christophe Dupre in the Yuste lab (Columbia University) (*Wicks et al., 1996*). The transgenic strain expressing GCaMP7b in endodermal epitheliomuscular cells was co-developed by Juliano Lab (University of California, Davis) and Robinson Lab (Rice University). Briefly, the plasmid with codon-optimized GCaMP7b under the EF1a promoter was constructed by GenScript (https://www.genscript.com). Injections were performed as previously described (*Rueden et al., 2017*) with the following modifications: (1) injection solution was prepared by mixing 1 µL 0.5% phenol red (Sigma P0290-100ML) with 6 µL plasmid DNA solution prior to centrifugation, and (2) embryos were fertilized for 1–2 hr prior to injection. Plasmid promoters were cloned in expression vector pHyVec2

(Addgene plasmid: #34790) using restriction sites NsiI. Plasmids were prepared by Maxiprep (Qiagen, Valencia, CA) and eluted in RNase-free water. A plasmid DNA solution of 1.4 µg/µL was injected into embryos using an Eppendorf FemtoJet 4x and Eppendorf InjectMan NI 2 microinjector (Eppendorf; Hamburg, Germany) under a Leica M165 C scope (Leica Microscopes, Inc; Buffalo Grove, IL). Viable hatchlings with mosaic expression were propagated by asexual reproduction, and asexual buds were screened and selected for increasing amounts of transgenic tissue until a line was established with uniform expression in the endodermal epithelial cells.

## Fluorescence imaging of *Hydra* nerve net

Distribution of neurons in the *Hydra* nerve net was fluorescently imaged with transgenic *Hydra vulgaris* expressing GFP (nGreen) in neurons and neuronal progenitors (*Figure 1a*; *Siebert et al., 2019*). *Hydra* was anesthetized with 0.05% chloretone and immobilized in an ~160 µm tall microfluidic chamber (*Badhiwala et al., 2018*). High-resolution fluorescence imaging was performed using a confocal microscope (Nikon TI Eclipse) and 10× (0.45 NA) objective, where the *Hydra* was imaged at a single plane with multiple fields of views stitched together to obtain an image of the whole animal (*Figure 1a*).

## *Hydra* resections

*Hydra* were placed in a Petri dish and covered with enough *Hydra* medium to prevent desiccation. When *Hydra* were relaxed and stationary, resections were performed with a single incision with a scalpel across the body. We referenced published images of in situ hybridizations of different cell types and spatial expression patterns to guide the location of incisions. For 'footless' *Hydra*, an axial cut above the peduncle removed approximately one third of the lower body, which included the peduncle and the basal disk. For 'headless' *Hydra*, an axial cut below the hypostome removed approximately one third of the upper body, including the tentacles and the hypostome. For 'bisected' *Hydra*, a transverse cut starting from the tip of the hypostome to the basal disk was made along the midline of the body. For 'body column' *Hydra*, an axial cut above the peduncle removed the lower body followed by another axial cut below the hypostome to remove the upper body region. This preparation resulted in an open tube body. *Hydra* were stored in an 18°C incubator after the excisions and until beginning the experiments. *Hydra* can seal open wounds within ~3–4 hr and repopulate the neuronal population to regain functionality in 30–72 hr (*Figure 3—figure supplement 1*, see 'Imaging regeneration of peduncle network'). To allow *Hydra* time to recover but not regain the functionality of lost neuronal cell types, we performed experiments 6–12 hr post amputation.

## Imaging regeneration of peduncle network

Transgenic *Hydra* (GcaMP6s, neurons) were axially cut in the middle of the body column to generate an oral and aboral halves (*Figure 3—figure supplement 1a*). The oral half was immobilized between two coverslips with an ~110 µm PDMS spacer. Calcium fluorescence was conducted for 20 min every 2 hr on Nikon TI Eclipse inverted microscope with 20% excitation light from Sola engine and GFP filter cube. We captured frames at ~10 Hz (100 ms exposures) with Andor Zyla 4.2 sCMOS camera with NIS software. We used 4× (0.2 NA) objective for wide-field imaging to fit the entire *Hydra* in the field of view reduce the likelihood of *Hydra* migrating out of the imaging frame. *Hydra* were exposed to blue excitation light for 20 min during imaging and remained under dark conditions for 100 min between subsequent imaging timepoints. One animal was imaged for ~20 hr starting with 1 hr post resection (*Figure 3—figure supplement 1b*). *Hydra* had a visibly open wound at the first imaging point but was not detectable after 3 hr. There were no visibly active neurons in the regenerating aboral end, indicating that the resected peduncle neurons had not regenerated. Another animal was imaged for ~30 hr starting with 37 hr post resection (*Figure 3—figure supplement 1c*). At the first imaging timepoint (t = 37 hr post resection), there were few neurons in the peduncle region that were active during body contractions, and these groups of neurons resembled a nerve ring as early as 41 hr post resection. Although the peduncle nerve ring seemed to have formed at this point, it appeared to not be as densely populated qualitatively as observed in whole animals.

## Microfluidic device fabrication

The mechanical stimulation devices are double-layer microfluidic devices with push-down valves custom-designed with CAD software (L-edit) and fabricated using standard photo- and soft-lithography techniques. All master molds were fabricated with transparency photomasks and SU-8 2075 (Micro-Chem). The master mold for the bottom *Hydra* layer (circular chambers, 3 mm diameter) was fabricated with the height of ~105–110 µm thick pattern (photoresist spun at 300 rpm for 20 s, 2100 rpm for 30 s). The master mold for the top valve layer (nine individual circular valves, 3 × 3 arrangement, 400 µm diameter each) was fabricated with height of ~110 µm thick (photoresist spun at 300 rpm for 20 s, 2100 rpm for 30 s). Polydimethylsiloxane (PDMS) Sylgard 184 was used to cast microfluidic devices from the master molds. The bottom *Hydra* layer (10:1 PDMS spun at 300 rpm for 40 s ~3 hr post mixing; cured for 12 min at 60°C) was bonded to the valve layer (~4 mm thick, 10:1 PDMS; cured for ~40 min at 60°C with holes punched for inlet ports) with oxygen plasma treatment (Harrick Plasma, 330 mTorr for 30 s) and baked for at least 10 min at 60°C. *Hydra* insertion ports were hole-punched through the two layers for *Hydra* layer with 1.5 mm biopsy punches, and the devices were permanently bonded ($O_2$ plasma treatment, 330 mTorr for 30 s) to 500 µm thick fused silica wafer (University Wafers) and baked for at least 1 hr at 60°C. The design files for the photomasks and step-by-step fabrication protocols will be available on https://www.openHydra.org (under Resource hub/Microfluidics).

*Hydra* were immobilized and removed from the microfluidic device using syringes to apply alternating positive and negative pressures as previously reported (*Badhiwala et al., 2018*). The microfluidic devices were reused after cleaning similarly to the protocol previously reported. Briefly, the devices were flushed with deionized water, sonicated (at least 10 min), boiled in deionized water (160°C for 1 hr), and oven-dried overnight.

After repeated use, the PDMS stiffness can change and affect the valve deflection and the actual force experienced by *Hydra* through the PDMS membrane. Additionally, uncontrollable conditions during the device fabrication process can also lead to small differences between devices. As a result, all data for *Figure 2* were taken with a single device and the response curve was used to calibrate (identify pressure that yielded ~60% response probability equivalent to 20–22 psi stimulus intensity) new devices.

## Microfluidic mechanical stimulation

We used compressed air to inflate the microfluidic valves. For temporal control over valve (on/off) dynamics, we used a USB-based controller for 24 solenoid pneumatic valves (Rafael Gómez-Sjöberg, Microfluidics Lab, Lawrence Berkeley National Laboratory, Berkeley, CA 94720) and a custom-built MATLAB GUI (available at https://www.openHydra.org) that allowed setting the stimulation parameters, such as the duration of valve 'on' (1 s), duration of valve 'off' (30 s), and the duration of stimulation (60 min, 119 cycles of valve 'on' and 'off'), and pre, post-stimulation acclimation/control period (20 min). We used a pressure regulator to manually control the air pressure into the valve manifold. In summary, we set the stimulation pressure with a pressure gauge to regulate the flow of air into the valve manifold. This valve manifold was controlled with a USB valve controller that allowed us to programmatically inflate the valve (turn it 'on') with pressurized air with custom stimulation parameters.

To test sensory motor response to mechanical stimuli, we used pressurized air to inflate the push-down valve and cause it to press down on the *Hydra* immobilized in the bottom layer. Each experimental condition had at least three *Hydra* each. For a given condition, replication experiments were conducted on different days. After an animal was immobilized inside the *Hydra* chamber, we selected one valve (from the nine valves over the entire chamber) that was directly above the mid-body column region to deliver stimuli. Although *Hydra* were free to move, we did not observe large displacement most of the time, and, as a result, the same valve remained in contact with the animal throughout the stimulation period.

We adjusted the air pressure using a pressure regulator for each experiment, and the valves were inflated using a USB valve controller (see above). The full-length stimulation experiment consisted of 20 min of no stimulation (control/acclimation) followed by 60 min of stimulation period (except habituation experiment where the stimulation period was 120 min), where valves were pulsed with constant pressure (0 [control], 5, 10, 15, 20, 22, or 25 psi) for 1 s every 31 s, then another 20 min of no

stimulation (control/acclimation). Shorter stimulation experiment (used for whole-animal muscle imaging of various resections) consisted of 20 min of no stimulation (acclimation/control) followed by 20 min of stimulation period (valves pulsed for 1 s every 31 s with a constant pressure of 0 or 22 psi). We chose a 20 min initial control period based on the high sensitivity to abrupt changes in light intensities (especially to blue wavelengths used for excitation of GCaMP) in *Hydra*, which leads to increased contractile activity for 2–5 min. Even with the stimulus repeated for 2 hr at a constant inter-stimulus interval, we found no obvious evidence of sensitization, habituation, or stimulus entrainment in *Hydra* (*Figure 2—figure supplement 9*). As a result, we chose not to randomize the inter-stimulus interval.

## Distribution of mechanical forces

We characterized the distribution of force exerted by a microfluidic valve by quantifying the movements of neurons due to mechanical stimulation. We performed fluorescence imaging in transgenic *Hydra* (nGreen) expressing GFP in neurons for ~8 min. We captured 8000 frames at ~16 Hz (50 ms exposures) with 4× objective (0.2 NA) and Andor Zyla 4.2 sCMOS camera with 2 × 2 binning (1024 × 1024 frame size) using MicroManager. *Hydra* was stimulated in the middle of the body column (valve pulsed for 1 s every 31 s, five times).

To quantify the movement of neurons and tissue throughout the *Hydra* body, we tracked a total of 222 neurons that were visible for 1 min capturing the first two stimulation trials. We performed semi-automated tracking with TrackMate plugin (ImageJ/Fiji) (*Rueden et al., 2017*; *Schindelin et al., 2012*; *Tinevez et al., 2017*) and manually corrected the tracks where neurons were misidentified. From these tracks, we calculated the displacement of each of the neurons between each frame (~50 ms). The average cellular displacement (0.4 µm per frame, 50 ms) was calculated by averaging the cellular displacements from all frames when the valve was not pressurized. We found a significantly increased displacement (6.2 µm, p<0.001) just after the valve was pressurized (and after the valve was depressurized 1 s later). We then generated a vector map of neuronal displacements by calculating the change in position of each of the neurons in the frame just after the valve was pressurized for the first stimulation trial (*Figure 2—figure supplement 1a*). We also plotted the cellular displacement for each of the neurons and the location of those neurons relative to the center of the valve. We averaged the highest three displacements in 50 µm radial band increments from the valve center to quantify how far the mechanical forces extended. This was a more conservative measurement as the neurons in different tissue layers (endodermal and ectodermal layers furthest from the valve) may have experienced different forces. By taking the average of the highest three displacements in 50 µm radial bands increments from the valve center, we identified the majority of the (shear) force was experienced by neurons bordering the valve in 250 µm radius.

## Fluorescence intensity from GFP *Hydra*

To prove that the fluorescence intensity changes we observed with calcium imaging are due to calcium activity and not motion artifacts from body contractions, we compared the average fluorescence intensities from three different transgenic *Hydra* lines: (1) expressing GFP pan-neuronally (nGreen line), (2) expressing GCaMP6s pan-neuronally, and (3) expressing GCaMP7b in ectodermal epitheliomuscular cells (*Figure 2—figure supplement 3*). For each *Hydra*, we compared the average fluorescence intensities from three different ROIs that included the peduncle region, whole-frame (for whole body) region, and valve region. We also measured the body length by taking the major axis of an ellipse fitted along the oral-aboral axis of the body column (from apex of the hypostome to the basal disc) after binarizing the fluorescence image.

For this analysis, we used fluorescence imaging from transgenic *Hydra* expressing GFP pan-neuronally (nGreen) performed in 'Distribution of mechanical forces.' *Hydra* was stimulated in the middle of the body column (valve pulsed for 1 s every 31 s, five times). We used calcium imaging from transgenic *Hydra* (expressing either GCaMP6s pan-neuronally or GCaMP7b in endodermal epitheliomuscular cells) stimulated in the middle of the body column (valve pulsed for 1 s every 31 s, 120 times). The average fluorescence intensity and body length traces were time-aligned to the onset of mechanical stimulation to show (1) changes in average fluorescence intensity in peduncle ROI and the whole-frame ROI are due to calcium activity (increase in fluorescence from in GCaMP lines) and

not motion artifact (no change in fluorescence from GFP line) and (2) increase or decrease in body length or average fluorescence from valve ROI are affected by stimulation artifacts.

## Average calcium fluorescence from large and small ROIs in the peduncle

We performed fluorescence imaging in transgenic *Hydra* expressing GCaMP6s in neurons for ~1 min during spontaneous behaviors (*Hydra* was not stimulated). We captured frames at ~16 Hz (50 ms exposures) with a 10× objective (0.45 NA) and Andor Zyla 4.2 sCMOS camera with 3 × 3 binning (682 × 682 frame size) using MicroManager.

Because calcium fluorescence decreases when neurons are inactive, tracking multiple neurons in a highly deformable region is a challenge. Nonetheless, we tracked nine neurons from the peduncle nerve ring that were visible (enough to track) for 1 min. We performed semi-automated tracking with TrackMate plugin (ImageJ/Fiji) (*Rueden et al., 2017*; *Schindelin et al., 2012*; *Tinevez et al., 2017*) and manually corrected the tracks where neurons were misidentified. We also used a large peduncle ROI, similar to how we used an ROI for quantifying neural response to stimulation in all other experiments. We calculated the average calcium fluorescence trace from small ROIs for individual neurons and large ROI for the peduncle region. Fluorescence intensity was normalized by calculating $\Delta F/F$, where $\Delta F = F\ F_0$ and $F_0$ is the minimum fluorescence from prior timepoints. The large peduncle ROI had highly correlated calcium fluorescence activity with smaller ROIs for individual neurons, which are known to belong to the contraction burst circuit (*Figure 2—figure supplement 2*). As a result, for all other experiments we use large peduncle ROI when measuring the neuronal activity.

## Whole-animal imaging of neural activity at different stimulus intensities

We characterized the neural response to repeated local mechanical stimuli at 0 (control), 5, 10, 15, 20, and 25 psi pressure with animals that expressed GCaMP6s in neurons (*Figure 2*). Each pressure condition was experimented with three animals on different days using the same device and stimulation protocol (valve pulsed for 1 s every 31 s for 60 min) to generate the pressure-response curves (total of 18 animals). Calcium fluorescence imaging for all experiments was conducted for the 100 min duration of the stimulation protocol (see Mechanical stimulation subsection) on Nikon TI Eclipse inverted microscope with 20% excitation light from Sola engine and GFP filter cube. We captured 100,000 frames at ~16 Hz (50 ms exposures) with Andor Zyla 4.2 sCMOS camera with 4 × 4 binning using MicroManager. For imaging neural activity, 12-bit low-noise camera dynamic range was used. To synchronize the stimulation onset times with imaging, we use used a data acquisition device (LabJack U3) to record the TTL frame out signal (fire-any, pin #2) from the Zyla and the valve on/off timestamps from the valve controller.

We used 4× (0.2 NA) objective for wide-field imaging to fit the entire *Hydra* in the field of view to reduce the likelihood of *Hydra* migrating out of the imaging frame. There are neurons in the oral region (hypostome) that are co-active with aboral region (peduncle) neurons during body contractions; however, neurons in the hypostome appeared to be much sparser than those in peduncle (*Video 18*). Discerning these hypostomal network neurons required higher magnification, which significantly reduced the field of view such that considerable amount of nervous tissue could move in and out of the imaging plane (z-plane) or frame (xy-plane),

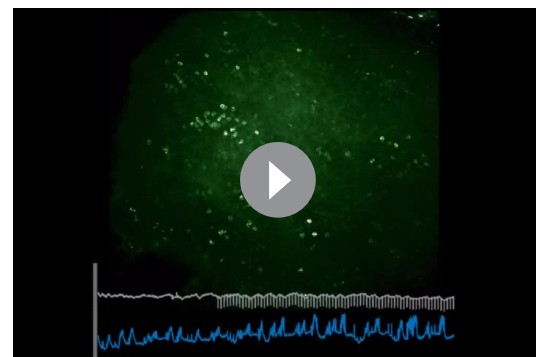

**Video 18.** Stimulated neural calcium activity in the hypostome and body column of normal animals. Dashed blue circle indicates the region of interest (ROI) used for calcium trace shown in blue (bottom). Dashed white circle indicates the location of the valve that presses down on *Hydra* when inflated. Gray trace shows the stimulus protocol, where vertical lines indicate valve 'on' times. Stimulus applied beginning t = ~20 min, valve is 'on' for 1 s and 'off' for 30 s. Imaged with 10 × 0.45 NA objective (playback 100×).
https://elifesciences.org/articles/64108#video18

making it difficult to obtain reliable calcium fluorescence time series.

## Whole-animal imaging of neural activity with different stimulation regions

To map the sensitivity of different body parts to mechanical stimuli, we performed calcium imaging of multiple animals (N = 3, whole animals expressing GCaMP6s in neurons) using imaging settings previously described with modified experimental protocol. Based on the range animal sizes (1–2.5 mm) and the size of the valve (400 μm), we stimulated *Hydra* body at three different regions along the oral-aboral axis: (1) the oral end (upper third of the body), (2) the aboral end (lower third of the body), and (3) the third near the midbody column. Each animal was stimulated at three different locations (20 min no stimulation, stimulation region #1 – 1 s every 31 s for 20 min, ~2 min no stimulation, stimulation region #2 – 1 s every 31 s for 20 min, ~2 min no stimulation, stimulation region #3 – 1 s every 31 s for 20 min using 22 psi). We randomized the order in which the three different body regions were stimulated in each of the three animals to avoid any stimulus entrainment artifacts. We then analyzed the peduncle nerve ring activity in response to mechanical stimulation of different body regions as detailed in 'Analysis of calcium activity' and 'Analysis of mechanosensory response'.

## Whole-animal imaging of neural activity in resected animals

For experiments that involved body lesions (*Figure 3—figure supplement 3*), the same experiment protocol and imaging settings as previously described (see 'Whole-animal imaging of neural activity at different stimulus intensities,' 20 min no stimulation, stimulation 1 s every 31 s for 60 min, 20 min no stimulation) were used with either 0 psi (control) or 20–22 psi (~60% response probability). Each condition group had three animals (total of 18 animals). The 'headless' and 'bisected' animals were prepared with the appropriate body regions removed as described previously ('*Hydra* resections'). Due to difficulty in tracking neurons and weak GCaMP expression in body column neurons, we were unable to quantify responses from 'footless' and 'body column' *Hydra* and thus excluded them from experiments.

## Simultaneous electrophysiology and calcium imaging of ectodermal epitheliomuscular cells

Electrical activity from the epitheliomuscular cells was measured simultaneously with calcium imaging of the ectodermal epitheliomuscular cells in transgenic *Hydra* using a nano-SPEARs device previously reported (*Figure 2—figure supplement 4*; *Badhiwala et al., 2018*). Briefly, transgenic *Hydra* expressing GCaMP6s in the ectodermal epitheliomuscular cells starved for at least 48 hr were used to measure the activity of the muscles (10 fps, 30 min, 4× objective with 15% light intensity). An inverted microscope with GFP filter and Andor Zyla 4.2 were used for capturing images. All electrical data was obtained with an Intan Technologies RHD2132 unipolar input amplifier (http://intantech.com) at a sampling rate of 1 kHz, low-frequency cutoff and DSP filter of 0.1 Hz and high-frequency cutoff of 7.5 kHz. From the calcium activity traces, we identified 30 s intervals of either high-amplitude activity or low-amplitude activity to perform cross-correlation analysis. The high- and low-amplitude activity regions were manually identified with a threshold of 20% of the highest peak in the calcium activity. The high-amplitude activity region occurred during contraction bursts. The low-amplitude activity region occurred during tentacle contractions for muscular activity imaging. Both the Intan amplifier and the Zyla were triggered with the same TTL signal. However, to account for any offset in the timing of the electrical and optical data, we measured the maximum of the cross-correlogram in a 50 ms (approximately one duty cycle of the trigger signal) window rather than the cross-correlation at zero offset to generate the correlation maps. For correlation map, each frame was down sampled to 64 × 64 pixels, and the fluorescence trace for the downsampled pixels across the 30 s intervals was cross-correlated with electrical activity during the same 30 s interval. The intensity of color in the correlation map was used to indicate correlation values.

## Whole-animal imaging of epitheliomuscular activity at different stimulus intensities

We characterized the muscle response to mechanical stimuli using animals that expressed GCaMP7b under the EF1a promoter in endodermal epitheliomuscular cells (*Figure 3*). We measured

contraction pulses and contraction bursts by averaging calcium activity in all the epitheliomuscular cells (*Figure 3a*). The correlation between peduncle nerve ring activity and muscle contractions has been previously established based on simultaneous electrophysiology and neuronal (*Badhiwala et al., 2018*) or ectodermal epitheliomuscular cells calcium imaging (*Figure 2—figure supplement 4*). The imaging protocol for epitheliomuscular cells was similar to the imaging of neural activity (see 'Whole-animal imaging of neural activity at different stimulus intensities'), except a 16-bit camera dynamic range was used to avoid saturating the sensor.

We first developed a partial psychometric curve for calcium activity of epitheliomuscular cells to confirm the dependence of response on stimulus intensity. Three animals were imaged for 20 min without stimulation and then stimulated at three different pressures (20 min at 15 psi, 20 min at 20 psi, 20 min at 25 psi; or in reverse order), and the epitheliomuscular response curve was used to identify the stimulus intensity (~22 psi), which yielded ~60% response probability (*Figure 2—figure supplement 5*). This stimulation intensity was also selected for stimulation of N = 8 whole animals (40 min of imaging; 20 min of no stimulation, 20 of stim for 1 s every 31 s with 22 psi) and animals with different body regions removed.

## Whole-animal imaging of epitheliomuscular activity in resected animals

The various body regions were removed as described previously ('*Hydra* resections') to prepare 'headless,' 'footless,' 'bisected,' and 'body column' *Hydra*. Note that this transgenic line of *Hydra* expressing GCaMP7b in the endodermal cells showed some signs of deficit. They were particularly sensitive to being handled and were more likely to dissociate after ~30 min in the chambers during long-term microfluidic immobilization and fluorescence imaging. The resected *Hydra* were especially difficult to image for the entire 100 min without any cell dissociation. This could be due to the specific promoter used for driving GCaMP expression or that phototoxicity is higher when there is a high expression of GCaMP in a larger number of cells. Because the lesioned animals were more likely to be damaged during microfluidic immobilization, the mechanical stimulation protocol (see Mechanical stimulation subsection) was shortened to total of 40 min of imaging with 20 min of no stimulation followed by 20 min of repeated stimulation (1 s every 31 s for 20 min). For three animals per each resection (total of 30 animals), we used the full-length protocol (100 min of imaging: 20 min of no stimulation, 60 min of stimulation every 31 s, and 20 min of no stimulation) to obtain higher statistical power for quantifying response probability (*Figure 3—figure supplement 3*). A total of 6–8 animals were stimulated for each resection condition (whole, N = 6 not stimulated, N = 8 stimulated; 'footless,' N = 4, not stimulated, N = 8 stimulated; 'headless,' N = 5 not stimulated, N = 8 stimulated; 'bisected,' N = 5 not stimulated, N = 8 stimulated; 'body column' N = 3 stimulated, N = 6 stimulated; total 61 animals, *Figure 3*; 30 of which were experimented with long [100 min] stimulation protocol; *Figure 3—figure supplement 3*).

## Analysis of calcium activity

To analyze neural response, we used the average fluorescence from the peduncle region. Tracking individual neural responses was difficult due to high deformability of the body and lack of fluorescence markers when neurons were not active; as a result, we looked at the synchronous firing activity of the neurons in the peduncle region, which is known to have high correlation with body contractions. To analyze epitheliomuscular response, we used average fluorescence from the whole animal to obtain the fluorescence signal over time and analyzed these signals similarly to neural response. Briefly, using ImageJ (Fiji) (*Rueden et al., 2017*; *Schindelin et al., 2012*) we used an ROI over the peduncle region or the whole *Hydra* to obtain fluorescence signal over time for neuronal or epitheliomuscular calcium activity, respectively.

From the fluorescence signal, we detected the large calcium spikes as individual contractions (contraction pulses or bursts) using MATLAB peak finding algorithm. We generated a raster plot of calcium spiking activity time-aligned with the stimulus where each row represented one stimulation trial with 15 s before and after stimulation onset. These raster plots were used to calculate the probability for spontaneous contraction and mechanosensory response.

For contractile behavior analysis, we then annotated the calcium spikes. Single-calcium spikes were labeled as single-pulse contraction. A volley of calcium spikes was labeled as a contraction burst. Both of these led to behavioral contractions, thus time between contractions was calculated

as the time between the end of a contraction event (single pulse or burst) and the start of the next one (as shown in *Figure 3—figure supplement 5*). Percent of contractions that are single-pulse contractions was calculated as the fraction behavioral contractions that were single-calcium spikes (not bursts). Contraction duration was used to indicate the amount of time contraction behavior lasted (time from rise in fluorescence signal to return to baseline, as shown in *Figure 3—figure supplement 5*).

## Analysis of mechanosensory response

To obtain the response probability (whether an animal had a contraction pulse – either a single pulse or a pulse from a burst) at the time of stimulation (when valve was pressurized), we generated a raster plot of fluorescence spikes time-aligned to stimulus onset and superimposed for each 30 s intervals (time between stimulus). We used an hour of activity (t = 20–80 min) to generate the raster plot and calculate response probability. We defined the 1 s window while the valve was pressed as the response window for each trial (*Figure 2—figure supplement 6*). We then calculated the fraction of all trials (119 trials over 60 min or 40 trials over 20 min) per animal that had at least one fluorescence spike ('contraction pulse') in the 1 s response window following stimulus onset to obtain the contraction probability. Extraction of raw fluorescence for ROIs was performed with ImageJ (Fiji) (*Rueden et al., 2017*; *Schindelin et al., 2012*) and postprocessing analysis was performed with MATLAB (using peak-finding algorithm to detect spikes). A one-way ANOVA with Bonferroni correction was used for statistical analysis.

For animals with shorter experiment duration, the mechanosensory response probability in stimulated animals was calculated over the 20 min segment (time = 20–40 min) with a fraction of all trials (40 trials over 20 min) that had at least one fluorescence spike. A one-way ANOVA with Bonferroni correction was used for statistical analysis when comparing multiple conditions. A paired t-test was used to compare the difference between probability of spontaneous contraction and mechanosensory probability in the same animals for each of the conditions. The effect size (magnitude of difference) was measured with Cohen's d and Cliff's delta.

## Analysis of spontaneous contraction

For non-stimulated animals, fluorescence activity from 1 hr of activity (time = 20–80 min) was converted into a raster plot with multiple (119 trials over 60 min) 30 s long intervals (to match the stimulation interval in stimulated animals). We obtained 'stimulation times' using a DAQ to record the signal from valve controller except the air pressure was set to 0 psi. The spontaneous contraction probability was calculated by taking the average of response probability from a random 1 s interval (*Figure 3* and *Figure 3—figure supplement 3*). Briefly, by sliding a 1 s window by ~0.3 s over the x-axis (30 s of stimulation interval) on the raster plot (pooled from all three animals for *Figure 4a*, per individual stimulated animal for *Figure 3—figure supplement 3*), we generated the distribution of the fraction of trials (119 trials × 3 animals over 60 min) with at least one contraction event in a random 1 s response window (*Figure 3—figure supplement 3*). The distributions were compared with a Kruskal–Wallis test.

For animals with shorter experiment duration, spontaneous contraction probability in stimulated animals was calculated similar to non-stimulated animals above, except the raster plot was generated with the first 20 min of fluorescence activity (time = 0–20 min) when no stimulation was applied. Briefly, we generated a distribution of random probabilities by sliding a 1 s window over the x-axis on the raster plot and used the distribution mean as the spontaneous contraction probability to compare with mechanosensory response probability from the same animal with a paired t-test (*Figure 3*). For non-stimulated animals with shorter experiment duration, spontaneous contraction probability was also calculated over the second 20 min interval (t = 20–40 min) to confirm the increase in contraction probability during mechanical stimulation was in fact due to stimuli and not just from temporal variation in spontaneous contractions (*Figure 3—figure supplement 4*). Note that these experiments were performed with *Hydra* constrained to ~110 µm thick microfluidic chambers. Although animals are able to behave under such confinements, the range of behavioral motifs and rates may be altered due to compression.

## Neuron subtype gene expression analysis with RT-qPCR

Resected *Hydra* were prepared as described above, with 12 polyps per biological replicate (except six whole polyps for control) and a total of two biological replicates per treatment. Approximately 12 hr post resections, tissue was frozen in 1 mL Trizol at −80°C until use. RNA was isolated using the Zymogen RNA Clean and Concentrator kit (Zymogen #R1017) with an in-column Zymogen DNAse I digestion (Zymogen #E1010) following the manufacturer's protocol. cDNA was synthesized using 1 µg of purified RNA and Promega M-MLV RNase H Minus Point Mutant Reverse Transcriptase (Promega, Madison, WI M3682) using the manufacturer's protocol for oligo dT-primed synthesis. cDNA synthesis was validated via PCR, and all cDNA samples were diluted 1:10 in nuclease-free water for use in qPCR experiments.

Each sample was run in three technical replicates per gene in a 10 µL qPCR reaction using Bio-Rad SsoAdvanced universal SYBR green master mix (Bio-Rad, Hercules, CA, 1725271). Samples were run on a CFX96 Touch Real-Time PCR Detection System (Bio-Rad 1855195). Data were analyzed with the $2^{-\Delta\Delta Ct}$ method using the 'tidyverse' package in R (*Livak and Schmittgen, 2001*; *R Development Core Team, 2020*).

Briefly, Cq values from technical replicates were pooled for subsequent analyses. *rp49* was used as an internal control to calculate ΔCq values after first being found to give similar results across all treatments when compared to a second control gene, *actin*. All results were normalized to the 'whole' animal samples. A template-free water control was performed for all primer sets to ensure contamination-free reactions. All qPCR primers (*Table 1*) were validated via a 10-fold serial dilution standard curve to have a binding efficiency over 90%.

## Whole-animal imaging with single-neuron resolution

We performed fluorescence imaging in transgenic *Hydra* (N = 3) expressing GCaMP6s in neurons for ~30 min (30 min of imaging: 20 min of no stimulation, 10 min of stimulation every 31 s). We captured frames at ~16 Hz (50 ms exposures) with a 10× objective (0.45 NA) and Andor Zyla 4.2 sCMOS camera with 3 × 3 binning (682 × 682 frame size) using MicroManager.

Because calcium fluorescence intensity is low (almost indiscernible from the autofluorescence of *Hydra*) when neurons are inactive, tracking multiple neurons in a highly deformable region without static nuclear fluorescence (such as RFP) is a challenge. Nonetheless, we manually tracked 5–6 neurons throughout the animal body that were visible (enough baseline fluorescence to track even when the neurons were inactive) for the entire duration of the imaging. We performed semi-automated tracking with TrackMate plugin (ImageJ/Fiji) (*Rueden et al., 2017*; *Schindelin et al., 2012*; *Tinevez et al., 2017*) to track individual neurons with small circular ROIs (~5 µm radius). We manually tracked (interpolating the current ROI location based on the past and future locations) the neurons when the fluorescence levels were dimmer than the background autofluorescence or when neurons were misidentified by TrackMate. We also used a large peduncle ROI, similar to how we used an ROI for quantifying neural response to stimulation in all other experiments. We calculated the average calcium fluorescence traces from small ROIs for individual neurons and large ROI for the peduncle region. Each of the fluorescence traces were corrected by calculating ΔF/F$_0$, where F$_0$ is the mean fluorescence intensity of the trace.

We performed the correlation analysis (MATLAB) of the calcium fluorescence time series for the individual neuron ROIs and peduncle ROI to identify groups of neurons with correlated activity. We calculated the correlation coefficients for the individual neurons and the peduncle ROI by correlating the entire 30 min calcium fluorescence trace for each of the neurons. These correlation coefficients were shown to be statistically significant then randomly reshuffled calcium fluorescence. We divided each of the neuronal calcium time series into 30 blocks. These individual blocks were randomly recombined to generate reshuffled time series for each of the neurons to calculate correlation coefficients. This random reshuffling of each neuron was repeated 1000 times to obtain the mean correlation coefficients for each of the neuron pairs (*Figure 4—figure supplement 3d–f*). We then calculated the z-score (x-µ)/σ, where x is the correlation coefficients of the original raw time series, µ is the mean correlation coefficients of the time series randomly reshuffled 1000 times, and σ is the standard deviation of the correlation coefficients of the time series randomly reshuffled (*Figure 4—figure supplement 3g–i*). To identify the neuronal clusters, we used a hierarchical clustering algorithm (MATLAB function 'linkage' with Euclidean distance of correlation coefficients) on the

**Table 1.** qPCR primers for neuronal subtypes.

| Subtype | transcriptID | F sequence | R sequence |
|---------|-------------|------------|------------|
| en1 | t29955aep | GCC GCA GTA TCA TCA TAC AAA TC | CCA TAA ACC ACA CAT CGC ATA AA |
| en2 | t20666aep | CTT CTT GCT TCT ATC CTC GTT CT | TAC CTC AAG TAA ATT ATC GGT CTC G |
| en3 | t33579aep | CGT TGG TAT GAC TAT AAT CGT TGT TAT G | AGG ATA CAT CAC CCA CCA AAT C |
| ec1A | t22316aep | GCC TTT CTT TAT CTC GGG TAT CT | ACC TCC CAT GAG TAG CTG TA |
| ec1B | t20807aep | AAG ATC TAC GAC GTC ATA TCA ATC A | TCA TGC CCT TAT TAC CCT CTT G |
| ec2 | t7411aep | GCG CCT TGT AAC TAT GGT CTT A | TTC GTA GAA CAT TGT CAT CTT CCT |
| ec3A | t9620aep | CGG TGC TGC TCC TAA TTC AA | CTC CGG TGC ACT GAT TTA TAG G |
| ec3B | t7664aep | TGT TTC AAA TGC AGA CGA AGA TG | GCG TGT TTA TTT GCC TGG AC |
| ec3C | t19558aep | CGG TTA GAT ACA CTG CGG TTA G | TAC GTG CCG TTC TTC GTT T |
| ec4 | t33899aep | GGC TTT AAT CGT TGT AGC TCT TG | CTT GCT ATC TTC TGA CAA GTG ATT G |
| ec5 | t6329aep | CCA ACA ATG GTC GAA TGA AGA AA | ATC GCC AGG TTT GTA TCC TTT A |
| RP49 | t6797aep | GCC AAA CTG GAG AAA ACC TAA AG | TCA GGC ATA AGA TGA CGT GTC |
| Actin | t11116aep | CGC CCT CGT AGT TGA TAA TGG | AAT CCT TCT GTC CCA TAC CAA C |

correlation coefficients. We then used leaf order from the linkage tree to sort the neurons and generate the correlation heat map.

We calculated the average fluorescence traces from the time-aligned calcium fluorescence to either spontaneous contractions or stimulus events to identify the roles of these neurons. For spontaneous contractions, the calcium fluorescence (t = 0–20 min) was time-aligned to the onset of contraction bursts or pulses from the peduncle ROI. For stimulated contractions, the calcium fluorescence (t = 20–30 min) was time-aligned to the onset of stimulus recorded with a DAQ (see 'Whole-animal imaging of neural activity at different stimulus intensities'). We further confirmed the different neuronal clusters identified by performing correlation analysis on all neurons from all three *Hydra* by calculating the correlation coefficients for the average fluorescence traces from the time-aligned calcium fluorescence to stimulus events (*Figure 4f*, g).

## Acknowledgements

We thank Dr. Rob Steele (UC Irvine), Dr. Christophe Dupre (Harvard University), and Dr. Rafael Yuste (Columbia University) for sharing transgenic *Hydra*. We also thank Dr. Guillaume Duret and Dr. Caleb Kemere for useful discussions, and Alondra Escobar for help with cell tracker. The promoter sequence for EF1a for generating transgenic *Hydra* was provided by Jack Cazet (UC Davis, Juliano lab), and the plasmid with GCAMP7b was cloned (and codon optimized) by GenScript. The open-source USB valve controller was built based on the design by Rafael Gómez-Sjöberg, Microfluidics Lab, Lawrence Berkeley National Laboratory, Berkeley, CA 94720. This work was supported in part by the NSF IOS-1829158 and National Institutes of Health in the United States R21AG067034. KNB is funded by training fellowships from the Keck Center of the Gulf Coast Consortia on the NSF Integrative Graduate Education and Research Traineeship (IGERT): Neuroengineering from Cells to Systems 1250104. This work was funded by NSF EDGE. KNB is funded by a training fellowship from the Keck Center of the Gulf Coast Consortia on the NSF IGERT: Neuroengineering from Cells to Systems 1250104. We also thank the Rice Shared Equipment Authority and nanofabrication cleanroom facility where the devices were fabricated.

## Additional information

### Funding

| Funder | Grant reference number | Author |
|---|---|---|
| National Science Foundation | IOS-1829158 | Celina E Juliano Jacob T Robinson |
| National Science Foundation | 1250104 | Krishna N Badhiwala |
| National Institutes of Health | R21AG067034 | Celina E Juliano Jacob T Robinson |

The funders had no role in study design, data collection and interpretation, or the decision to submit the work for publication.

### Author contributions

Krishna N Badhiwala, Conceptualization, Data curation, Software, Formal analysis, Validation, Investigation, Visualization, Methodology, Writing - original draft, Writing - review and editing; Abby S Primack, Methodology, Writing - review and editing, Developed transgenic hydra model; Celina E Juliano, Writing - review and editing; Jacob T Robinson, Conceptualization, Supervision, Funding acquisition, Writing - original draft, Writing - review and editing

### Author ORCIDs

Krishna N Badhiwala (iD) https://orcid.org/0000-0002-7180-5047
Celina E Juliano (iD) https://orcid.org/0000-0003-4222-0987
Jacob T Robinson (iD) https://orcid.org/0000-0002-3509-3054

### Decision letter and Author response

Decision letter https://doi.org/10.7554/eLife.64108.sa1
Author response https://doi.org/10.7554/eLife.64108.sa2

## Additional files

### Supplementary files

- Transparent reporting form

### Data availability

Source data for Figure 2, Figure 3 and figure 4 have been provided. We are preparing our raw data (microscopy images) to share in the most useful format for all figures and will make these datasets available on generic databases/institutional repository.

The following dataset was generated:

| Author(s) | Year | Dataset title | Dataset URL | Database and Identifier |
|---|---|---|---|---|
| Badhiwala KN | 2021 | Mechanosensation Hydra | https://doi.org/10.6084/m9.figshare.c.5490159.v1 | figshare, 10.6084/m9.figshare.c.5490159.v1 |

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
