## [Decision Letter]

**Acceptance summary:**

The reviewers are satisfied with the thought and effort the authors put into this revised manuscript and are now happy to publish this interesting study without reservation. In this emerging model organism for neuroscience, the authors nicely demonstrate with calcium imaging and loss-of-function approaches, the identification of diverse sensorimotor activity patterns to mechanical stimulation, driven by unique networks of neurons.

**Decision letter after peer review:**

[Editors’ note: the authors submitted for reconsideration following the decision after peer review. What follows is the decision letter after the first round of review.]

Thank you for submitting your work entitled "Multiple nerve rings coordinate *Hydra* mechanosensory behavior" for consideration by *eLife*. Your article has been reviewed by 3 peer reviewers, one of whom is a member of our Board of Reviewing Editors, and the evaluation has been overseen by a Senior Editor. The reviewers have opted to remain anonymous.

We are sorry to say that, after consultation with the reviewers, we have decided that your work will not be considered further for publication in *eLife*.

Specifically, all of the reviewers agreed that the emerging Hydra system holds great promise for neuroscience discoveries. Moreover, some of the findings presented here have the potential to be of use to other scientists who work in this system. However, we felt that the findings here were too preliminary and underdeveloped to warrant publication in *eLife* in its current form. In particular reviewers felt that (1) multiple locations across the Hydra's body should be stimulated coupled with mapping the behavioral and neuronal correlates of such stimulation, (2) the pan-neuronal nature of the bulk calcium measurements made it challenging to fully appreciate which neuronal circuits might be driving the sensorimotor responses, (3) uniform proxies for measuring/plotting the behavior would be useful, (4) the ablation studies lacked cellular resolution, similar to the calcium imaging experiments. Below, please find detailed comments that we hope will help to strengthen this work for submission elsewhere.

*Reviewer #1:*

The manuscript by Badhiwala et al., is an interesting study using the emerging model system Hydra, which has many advantages for studying the entire nervous system of an animal during simple behavior. Some of the foundational neuroscience papers in this field have only come out in the past few years, and new studies such as the one here, have the potential to contribute to an important early literature. Despite clear reasons for enthusiasm, there are a few issues in this work that require attention prior to publication. Although I appreciate building the microfluidic device with simultaneous pan-neuronal imaging, the nature of the new biological insights provided here are somewhat limited. The preliminary nature of some experiments also make it challenging to judge the potential implications of the work.

1. The pressurized stimulation of the hydra appeared to be specific to the center of the body. The authors don't mention why this region was chosen, which seems critical to this study. Relatedly, it appears important to test multiple areas across the hydra with this system. Might we expect to see different sensorimotor behaviors, and thus different neural outputs?

2. The authors reference a recent single cell study characterizing multiple neuronal cell types in hydra. This work would greatly benefit by using some cell-type resolution studies to determine the functional nature of the neurons being activated as opposed to solely using pan-neuronal GCAMP imaging. It appears that since they can put GCAMP in all neurons, they may be able to also introduce GCaMP into specific subsets of neurons based on cellular identity. This point becomes more salient because a major take-home from this paper is that the spontaneous behavior and firing patterns is nearly identical to the stimulus evoked patterns, except for an apparent increase in firing rate. The true nature of the mechanosensory response might be revealed with cell-type specific experiments.

3. Although the authors reference whole animal imaging, they focus imaging analysis on peduncle and hypostomal nerve rings, despite the videos showing calcium activity in other areas throughout the body. Moreover, are the authors certain their pan-neuronal genetic strategy equally samples neurons throughout the body? In other words, is it possible that the apparent increase in activity in the nerve ring over other areas is being driven by a technical artifact of these neurons being labeled better?

4. While I appreciate the resection studies to get at "loss-of-function" experiments, this is a somewhat blunt hammer approach, and potentially confounds clear interpretation. It is important to establish which neurons are killed and to what extent, and how many, if any began to regenerate throughout this process. The concern is that residual activity from neurons not ablated may be driving this response, especially in light of the authors surprising finding that "footless" animals show that the aboral nerve ring is not required for spontaneous or mechanosensory responses.

*Reviewer #2:*

The Hydra, in the phylum cnidaria, is a near microscopic freshwater animal that has recently resurfaced as an attractive model organism in neuroscience due to its optically accessible transparent body, sparsely distributed neural network, and simple behaviors. In this manuscript, Badhiwala and colleagues use calcium imaging of the Hydra neural network, combined with surgical resection and microfluidics pressure stimulation to identify body regions indispensable for mechanosensory activity. They report that while resection of the aboral region did not abolish the mechanical response, resection of the oral region attenuated this response, while combined resection of oral and aboral regions showed the greatest effect. They also find a correlation between reduced stimulated activity and spontaneous activity, suggesting a common mechanism that gives rise to both activities. While this study takes on an innovative approach by using a microfluidics device to mechanically stimulate the hydra under optical recording there are a number of conceptual and technical limitations. Perhaps my biggest concern is that despite the potential, the data are rather low resolution (body transections and bulk calcium responses) and as such the conclusions supported by the provided data do not significantly extend our current knowledge.

1. The authors have designed a microfluidic device that allows them to simultaneously mechanically stimulate, monitor movement and functionally image a hydra. The highly quantifiable nature of the microfluidic device a great asset, although this potential is not deeply explored. While I can see how the microfluidic stimulation could offer benefits over fluid jet or blunt probe, more in-depth characterization is needed.

2. What is the spatial distribution of the pressure pulse stimulus on the Hydra body? How far does the mechanical force spread from the region directly touching the pressure valve?

3. The use of the microfluidic device was limited. Have the authors attempted to map mechanical sensitivity across the Hydra body by stimulating different sites?

4. The authors spatially average a population response from a large region of interest as compared to recording calcium responses from single neurons. This should be specifically stated in the Results section. More importantly, to provide insight into network function much smaller ROIs over multiple sites are needed instead of the bulk activity of the entire peduncle. This analysis would allow the authors to significantly extend the findings from this work, as the lure of the optically clear and small hydra is that neural representation and coding can be tracked over large portions of the network at cellular resolution.

5. It is unclear where the recorded signals are coming from and if movement is creating artifacts. Have authors corrected for movement? The supplemental movies show a stationary region of interest and moving animal, in some cases parts of animal moving in and out. Furthermore, please explain how the background is subtracted. There is a large fluorescent signal coming from of the entire body/ middle columnar part of the body and spontaneous firing that makes interpretation of the data difficult.

6. Contraction is a behavioral response of the animal; however, the authors use 'contraction' do describe calcium imaging responses throughout the figures and text. This should be avoided.

7. The authors may wish to rephrase the wording of the paper title. I do not think this work establishes definitively that "multiple nerve rings" are important for coordinating mechanosensory behavior.

8. Furthermore, the claim that the observed "linear relationship" between the spontaneous contraction probability and resection type is evidence for shared neural pathways is not strongly supported by the provided data. These data are fairly coarse resolution and include only 3 animals in each group with highly variable responses (Figure 4C). Additionally, they do not provide evidence to distinguish the motor circuits they hypothesized these neural nets converge upon.

*Reviewer #3:*

The authors probe mechanosensory processing in Hydra by measuring calcium activity in neurons and muscles in response to precise mechanosensory stimulation in whole and resected animals. The authors' claims are well supported by the evidence. The development of a mechanosensory delivery system for Hydra is also a significant methodological advancement. Taken together, the work advances our understanding of the Hydra nervous system and is a needed step towards developing Hydra as a powerful model for systems neuroscience.

1) One issue is that different measures of "mechanosensory response" are used at different places in the manuscript. In some contexts, a response is defined as calcium activity in neurons (Figure 2), and elsewhere as calcium activity in muscles (Figure 3 and 4). And in Figure 2 SuppFig2 muscle contractions are also measured using MeKs. The relation between neural activity, muscle activity and body movement is, of course, of high interest, and the paper explores this. But, if technically possible, it would be helpful to report a single metric of behavior that could be used in all experiments. For example, it might be possible to use video of the animal's pose or body length to measure contractions in all experiments. At a minimum the reasoning behind choice of measurement of response for each experiment could be discussed explicitly.

2) Related: Without a consistent measure of behavior, it will be important to further clarify figures so that a reader can tell at-a-glance how contraction probability is being measured.

---

## [Author Response]

[Editors’ note: The authors appealed the original decision. What follows is the authors’ response to the first round of review.]

Specifically, all of the reviewers agreed that the emerging Hydra system holds great promise for neuroscience discoveries. Moreover, some of the findings presented here have the potential to be of use to other scientists who work in this system. However, we felt that the findings here were too preliminary and underdeveloped to warrant publication in eLife in its current form. In particular reviewers felt that (1) multiple locations across the Hydra's body should be stimulated coupled with mapping the behavioral and neuronal correlates of such stimulation, (2) the pan-neuronal nature of the bulk calcium measurements made it challenging to fully appreciate which neuronal circuits might be driving the sensorimotor responses, (3) uniform proxies for measuring/plotting the behavior would be useful, (4) the ablation studies lacked cellular resolution, similar to the calcium imaging experiments.

Based on these comments and the comments from the individual reviewers, we have performed new experiments that more comprehensively test our hypothesis that distinct neuronal networks coordinate the mechanosensory response in *Hydra*.

Specifically, we performed single-cell imaging of several neurons during spontaneous and stimulated contractions (Experiment E2) in three animals to identify a group of “mechanically responsive” neurons.

In addition, we address the enumerated questions as follows:

1. We performed stimulation at multiple locations on the *Hydra* body and quantified behavioral and neural responses to see if the location of stimuli affects the animal’s response. (Experiment E1).

2. As described above, we performed single-cell-resolved calcium imaging to determine the differences in the pattern of neural activity between spontaneous and stimulated contraction (Experiment E2).

3. We provide a more explicit explanation for the different measures of behavior (Analysis A3).

4. We performed qPCR to quantify how well our resection experiments remove the targeted cell types (similar to how one would quantify a genetic knock down) (Experiment E3).

We also request that the editors and reviewers recognize that *Hydra* is a model system that is not as well developed as other organisms. Thus, many experiments like cell-type specific labeling that may be relatively routine in organisms like *C. elegans* and *Drosophila* are simply not yet possible given the available techniques. As a result, we turn to non-traditional methods like surgical resectioning and large ROI imaging, which are unique solutions given the constraints of our model system. We hope that the editors and reviewers recognize these unique challenges associated with non-traditional model organisms and consider that these are the data available to those of us working in these lesser developed systems.

List of additional experiments and analysis:

Experiments:

1. Experiment E1 (Reviewer 1,2,3): Mapping mechanical sensitivity/response across the *Hydra* body.

We thank the reviewers for this suggestion to map mechanical sensitivity across the *Hydra* body. We performed new experiments to test *Hydra*’s response to stimulation along the body. Unlike in *C. elegans,* where the location of stimulation leads to different behavioral responses, we find the same aversive contractile response regardless of where along the body *Hydra* is stimulated. However, we did observe differences in response probability (sensitivity) depending on where the animal was stimulated. Specifically, we found the aboral region of *Hydra* to be less sensitive to mechanical stimulation than the center of the body.

We describe the new experiment with this new text in the methods:

“Whole-animal imaging of neural activity with different stimulation regions

To map the sensitivity of different body parts to mechanical stimuli, we performed calcium imaging of multiple animals (N = 3, whole animals expressing GCaMP6s in neurons) using imaging settings previously described with modified experimental protocol. Based on the range animal sizes (1-2.5 mm) and the size of the valve (400 um), we stimulated Hydra body at three different regions along the oral-aboral axis: (1) the oral end (upper third of the body), (2) the aboral end (lower third of the body), and (3) the third near the mid-body column. Each animal was stimulated at three different locations (20 min no stimulation, stimulation region #1 – 1 sec every 31 sec for 20 min, ~2 min no stimulation, stimulation region #2 – 1 sec every 31 sec for 20 min, ~2 min no stimulation, stimulation region #3 – 1 sec every 31 sec for 20 min using 22 psi). We randomized the order in which the three different body regions were stimulated in each of the three animals to avoid any stimulus entrainment artifacts. We then analyzed the peduncle nerve ring activity in response to mechanical stimulation of different body regions as details in “Analysis of calcium activity” and “Analysis of mechanosensory response”.”

We have added this new section to the results:

“Hydra sensitivity to mechanical stimuli is lowest near the aboral end

Because of the diffuse neural architecture of Hydra, we expected each patch of Hydra tissue to be equally responsive to mechanical stimuli. […] The only difference we observed was the fact that the response probability depended on where along the oral and aboral axis we delivered the mechanical stimulus.”

2. Experiment E2 (Reviewer 1, 2): Single cell resolution of neuronal activity.

We agree with the reviewers that this work would greatly benefit from some single cell-type resolution study. In theory, we should be able to take advantage of the optical transparency of the animal to track individual neurons or drive the expression of GCaMP in specific subsets of neurons based on the cell-type specific biomarkers. However, we currently do not have these transgenic lines and generating these transgenic *Hydra* lines would take more than a year.

While we are unable to automatically track individual neurons due to the lack of nuclear fluorescent reporter (i.e., ratiometric imaging with RFP), we were able to manually track several neurons that were visible for the entirety of the experiment (30 minutes) in three different *Hydra*.

By analyzing these data, we have made two new findings that we found surprising:

1. There are a group of neurons whose pattern of activity is qualitatively different from neurons that belong to contraction burst or rhythmic potential networks. We refer to these neurons as ‘mechanically responsive’ (MR) neurons because these neurons showed a clear response following mechanical stimulus but not spontaneous contractions. The CB network showed very similar activity for both stimulated and spontaneous contractions. Together, these data show a unique neural program for simulated vs spontaneous contractions.

2. The MR neurons showed a significantly delayed response to mechanical stimulation responding approximately 10 seconds after a stimulus evoked contraction. These data suggest that these MR neurons (found primarily in the oral network) may have a separate activation pathway that parallels the activation of the CB neurons and plays a unique role in stimulated contractions compared to spontaneous contractions. This is the first evidence of parallel sensory processing in *Hydra* and was not expected from our initial data.

We have included these new results and a discussion of how they advance our knowledge on pgs. 15-17. We thank the reviewers for suggesting we take this closer look – we were surprised by what we found!

We describe the new experiment with this new text in methods:

“Whole animal imaging with single neuron resolution

We performed fluorescence imaging in transgenic Hydra (N = 3) expressing GCaMP6s in neurons for ~30 min (30 min of imaging: 20 min of no stimulation, 10 min of stimulation every 31 sec). We captured frames at ~16Hz (50 msec exposures) with a 10x objective (0.45 N.A.) and Andor Zyla 4.2 sCMOS camera with 3x3 binning (682x682 frame size) using MicroManager. […] We calculated the average fluorescence traces from the time-aligned calcium fluorescence to either spontaneous contractions or stimulus events to identify the roles of these neurons. For spontaneous contractions, the calcium fluorescence (t = 0-20 min) was time-aligned to the onset of contraction bursts or pulses from the peduncle ROI. For stimulated contractions, the calcium fluorescence (t = 20-30 min) was time-aligned to the onset of stimulus recorded with a DAQ (see “Whole-animal imaging of neural activity at different stimulus intensities”). We further confirmed the different neuronal clusters identified by performing correlation analysis on all neurons from all three Hydra by calculating the correlation coefficients for the average fluorescence traces from the time-aligned calcium fluorescence to stimulus events (Figure 4f.g).”

We have added this new Results section to the main text: “Oral and aboral networks show different patterns of activity during spontaneous contractions compared to mechanically stimulated contractions”

And we have added three new supplementary figures (Figure 4 —figure supplement 1-3) to present results from biological replicates for single cell resolution of neuronal activity and show statistical significance of our correlation analysis.

3. Experiment E3 (Reviewer 1 and 2): cell-type resolution studies to determine the functional nature of the neurons.

We agree with the reviewers that this work would greatly benefit from some single cell-type manipulation study and in theory, we should be able to genetically modify specific subsets of neurons based on the cell-type specific biomarkers to inhibit or excite specific neurons to study their function. However, we currently do not have these transgenic lines and generating these transgenic *Hydra* lines may take more than a year since we may have to test several promoters.

While we cannot at this time perform cell-type specific optogenetic experiments, we can take advantage of the spatial distribution of neuronal subtypes along the oral-aboral axis in *Hydra*. By resecting specific body regions, we can remove specific neuronal subtypes. We have now added an experiment to the manuscript to demonstrate that we were successful in removing specific neuron subtypes. We performed quantitative PCR to measure the expression levels of subtype specific neuronal marker genes in differently resected animals. We found evidence of a complete loss of the predicted specific neuronal subtypes 12 hours after the resections were performed. These findings support body resections as an alternative technique for cell-type specific functional studies (like optogenetics) and confirm previous findings that neurons are not regenerated within 12 hours of their removal.

We describe this new experiment with new text in methods:

“Neuron subtype gene expression analysis with RT-qPCR

Resected Hydra were prepared as described above, with 12 polyps per biological replicate (except 6 whole polyps for control) and a total of two biological replicates per treatment. Approximately 12 hrs post resections, tissue was frozen in 1 mL Trizol at -80°C until use. RNA was isolated using the Zymogen RNA Clean and Concentrator kit (Zymogen #R1017) with an in-column Zymogen DNAse I digestion (Zymogen #E1010) following the manufacturer’s protocol. cDNA was synthesized using 1 µg of purified RNA and Promega M-MLV RNase H Minus Point Mutant Reverse Transcriptase (Promega, Madison, WI M3682) using the manufacturer’s protocol for oligo dT-primed synthesis. cDNA synthesis was validated via PCR and all cDNA samples were diluted 1:10 in nuclease free water for use in qPCR experiments.

Each sample was run in three technical replicates per gene in a 10 uL qPCR reaction using Bio-Rad SsoAdvanced universal SYBR green master mix (Bio-Rad, Hercules, CA 1725271). Samples were run on a CFX96 Touch Real-Time PCR Detection System (Bio-Rad 1855195). Data were analyzed with the 2^-ΔΔCt method using the tidyverse package in R72.

Briefly, Cq values from technical replicates were pooled for subsequent analyses. rp49 was used as an internal control to calculate ΔCq values after first being found to give similar results across all treatments when compared to a second control gene, actin. All results were normalized to the “whole” animal samples. A template-free water control was performed for all primer sets to ensure contamination-free reactions. All qPCR primers (Table 1) were validated via a 10-fold serial dilution standard curve to have a binding efficiency over 90%.”

We have added the following new text to reflect the new findings in the main text:

“We confirmed that 6-12 hours after resection the animals indeed showed loss of specific neuronal cell types by measuring the expression levels of subtype-specific neuronal markers via qPCR (Figure 3 —figure supplement 2).”

“We identified at least two neuronal networks (MR and CB network) with distinct neuronal activities mediating the stimulated responses (Figure 4), where the CB network of neurons show fast calcium responses, and the MR neurons show slower calcium response. It is possible that the MR neurons, primarily found in the oral end (N = 3 Hydra), may consist of ec4 neurons, which are the only neuronal subtype in the oral end with an unknown function. The other orally located ectodermal neuron populations, ec1B and ec3C, are suggested to belong to the CB and RP1 circuits, respectively^48^.”

We have added a new figure (Figure 3 —figure supplement 2)

5. Experiment E5 (Reviewer 3): GFP control to show fluorescence intensity is due to calcium activity and not due to artifacts of the contraction.

To prove the fluorescence intensity is due to calcium activity and not an artifact of the body contractions, we performed a short experiment where we stimulated *Hydra* mechanically (1 sec every 31 sec, 5 times). By comparing the GFP intensity with GCaMP intensity from various regions of interest, we confirmed the increase in fluorescence intensity in the way we analyzed our calcium imaging (using large ROIs) was not due to motion artifacts.

We describe the new experiment with new text in the methods:

“Fluorescence intensity from GFP Hydra

To prove the fluorescence intensity changes we observed with calcium imaging are due to calcium activity and not motion artifacts from body contractions, we compared the average fluorescence intensities from three different transgenic Hydra lines: (1) expressing GFP pan-neuronally (nGreen line), (2) expressing GCaMP6s pan-neuronally and 3) expressing GCaMP7b in endodermal epitheliomuscular cells (Figure 2 —figure supplement (3). […] The average fluorescence intensity and body length traces were time-aligned to the onset of mechanical stimulation to show (1) changes in average fluorescence intensity in peduncle ROI and the whole frame ROI are due to calcium activity (increase in fluorescence from in GCaMP lines) and not motion artifact (no change in fluorescence from GFP line) and (2) increase or decrease in body length or average fluorescence from valve ROI are affected by stimulation artifacts.”

We have added new text to the main text:

“We further confirmed that this signal is not the result of motion artifacts by measuring fluorescence from Hydra (nGreen) that express GFP pan-neuronally using a similar ROI. In that case we do not see the strong fluorescence signals associated with contraction pulses and bursts (Figure 2 —figure supplement 3e-f).”

We have added new supplementary Figure (Figure 2 —figure supplement 3):

Analysis:

1. Analysis A1 (Reviewer 2): Spatial distribution of the mechanical force.

We thank the reviewer for suggesting further characterization of the distribution of mechanical forces from the microfluidic valves. Briefly, we performed additional experiments with *Hydra* expressing GFP in neurons and individually tracked the movements of neurons semi-automatically. We mapped the cellular displacement from a total of 222 neurons when the valve is compressed to illustrate the spatial distribution of mechanical force.

We describe this new experiment with new text in methods:

“Distribution of mechanical forces

We characterized the distribution of force exerted by a microfluidic valve by quantifying the movements of neurons due to mechanical stimulation. […] By taking the average of the highest three displacements in 50 µm radial bands increments from the valve center, we identified the majority of the (shear) force was experienced by neurons bordering the valve in 250 µm radius.”

We have added this new paragraph to the main text:

“Experiments showed that this stimulation paradigm delivered a local mechanical stimulation with most of the mechanical force localized to a radius of approximately 250 µm around the microfluidic valve. […] This lateral displacement decreased for neurons that were farther from the center of the valve. Neurons more than 750 µm from the microfluidic valve center showed a negligible displacement of less than 5 µm (95% CI lower bound = 5.8 µm), which is ~550% less than the displacement of neurons bordering the valve.”

We have added a new figure (Figure 2 —figure supplement 1):

2. Analysis A2 (Reviewer 2): low-throughput (N=3) Figure 4C does not strongly support shared neural pathways.

We thank the reviewer for this feedback and agree these results do not strongly support shared neural pathways. As a result, we have decided to replace this figure with analysis of calcium activity from individual neurons in the oral and aboral regions during spontaneous and stimulated behaviors, which we believe do a better job of discriminating the roles of the oral and aboral network in spontaneous and stimulated contractions. See Experiment E2 above for more details.

We have replaced figure 4 with New Figure 4 and added three new supplement figures.

3. Analysis A3 (Reviewer 3): Single metric for behavioral response.

We agree a single metric for behavioral response using body length or posture would be an excellent way to compare across different conditions and experiments. However, it is extremely challenging to completely automate postural tracking (automation is required because this manuscript includes calcium imaging from >100 animals) in highly deforming animals from calcium fluorescence images.

To illustrate the performance of posture tracking with calcium fluorescence images, we have attempted this analysis in three different transgenic animals: expressing GFP in neurons (nGreen), GCaMP6s in neurons and GCaMP7b in endodermal epitheliomuscular cells. We quantified the change in body length during mechanical stimulation and found that the body length was not a reliable metric for behavioral response due to stimulation artifacts. Specifically, stimulation caused an increase in body length (or size) suggesting body elongation which masked the expected body contractions. However, the calcium imaging provided a good metric for assessing contraction pulses and bursts (Figure 2)

We have added the following new text:

“Body length proved to be an unreliable quantification of contractions due to the stimulation artifacts (Figure 2 —figure supplement 3); however, we were able to accurately measure muscle activity associated with contractions by imaging calcium spikes in the epithelial muscle cells (Figure 2 —figure supplement 4, 5 and Supp. Movie 4, 5)”

We have added new supplementary Figure (see Figure 2 —figure supplement 3):

Reviewer #1:The manuscript by Badhiwala et al. is an interesting study using the emerging model system Hydra, which has many advantages for studying the entire nervous system of an animal during simple behavior. Some of the foundational neuroscience papers in this field have only come out in the past few years, and new studies such as the one here, have the potential to contribute to an important early literature. Despite clear reasons for enthusiasm, there are a few issues in this work that require attention prior to publication. Although I appreciate building the microfluidic device with simultaneous pan-neuronal imaging, the nature of the new biological insights provided here are somewhat limited. The preliminary nature of some experiments also make it challenging to judge the potential implications of the work.

We thank the reviewer for recognizing the potential of our work. We believe our responses to the individual points below will clarify the significance of our contributions by providing additional biological insights about mechanical sensitivity in *Hydra* and support for our methodologies.

1. The pressurized stimulation of the hydra appeared to be specific to the center of the body. The authors don't mention why this region was chosen, which seems critical to this study. Relatedly, it appears important to test multiple areas across the hydra with this system. Might we expect to see different sensorimotor behaviors, and thus different neural outputs?

We chose to stimulate the body column because that region is the least likely to move away from the stimulation valve, while oral and aboral extremities tend to exhibit a lot of deformations and movements.

We previously provided explanation for the selection of stimulation region in the methods:

“After an animal was immobilized inside the Hydra chamber, we identified one valve (from the nine valves over the entire chamber) that was directly above the mid-body column region and that valve was selected for delivering stimuli. Although Hydra were free to move, we did not observe large displacement most of the time and as a result, the same valve remained in contact with the animal throughout the stimulation period.”:

We have now added this new text to main text for why this region (center of the body/body column) was chosen as the stimulation site for all experiments:

“We selected the middle of the body for stimulation region to help ensure that we stimulated roughly the same region of the animal throughout each experiment. This choice was based on the observation that the body column region was relatively stationary, whereas the oral and aboral extremities had large displacements during body contractions and elongations.”

We agree with the reviewer that it is important to test multiple areas across the *Hydra* body. We have added a new experiment (Experiment E1, see above) to stimulate *Hydra* at different regions along its body. We found the same aversive contractile response regardless of where along the body *Hydra* is stimulated. However, we did observe differences in response probability (sensitivity) depending on where the animal was stimulated. Specifically, we found the aboral region of *Hydra* to be less sensitive to mechanical stimulation than the center of the body.

[see Experiment E1 above for more details]

2. The authors reference a recent single cell study characterizing multiple neuronal cell types in hydra. This work would greatly benefit by using some cell-type resolution studies to determine the functional nature of the neurons being activated as opposed to solely using pan-neuronal GCAMP imaging. It appears that since they can put GCAMP in all neurons, they may be able to also introduce GCaMP into specific subsets of neurons based on cellular identity. This point becomes more salient because a major take-home from this paper is that the spontaneous behavior and firing patterns is nearly identical to the stimulus evoked patterns, except for an apparent increase in firing rate. The true nature of the mechanosensory response might be revealed with cell-type specific experiments.

We agree with the reviewers that this work would greatly benefit from some single cell-type resolution study. In theory, we should be able to take advantage of the optical transparency of the animal to track individual neurons or drive the expression of GCaMP in specific subsets of neurons based on the cell-type specific biomarkers. However, we currently do not have these transgenic lines and generating these transgenic *Hydra* lines may take more than a year since we may have to test several promoters.

While we are unable to automatically track individual neurons due to the lack of nuclear fluorescent reporter (i.e. ratiometric imaging with RFP), we manually tracked several neurons that were visible for the entirety of the experiment (20 minutes) in new experiment (Experiment E2, see above).

Although both spontaneous and stimulated behaviors do in fact converge onto the contraction burst network, we found distinct patterns of activity for spontaneous and stimulated behaviors that suggests different networks perform different computations for spontaneous and stimulated behaviors.

[see Experiment E2 above for more details]

3. Although the authors reference whole animal imaging, they focus imaging analysis on peduncle and hypostomal nerve rings, despite the videos showing calcium activity in other areas throughout the body. Moreover, are the authors certain their pan-neuronal genetic strategy equally samples neurons throughout the body? In other words, is it possible that the apparent increase in activity in the nerve ring over other areas is being driven by a technical artifact of these neurons being labeled better?

We thank the reviewer for suggesting that we establish that the calcium dynamics are not an artifact of movement or differences in neural density. To address this point, we performed experiments (Experiment E5, see above) with a line that expresses GFP in all neurons. When we compared these experiments to experiments in GCaMP animals, we found that the Ca++ activity cannot be explained by movement or labeling artifacts. These data are also consistent with previous experiments that imaged calcium activity in *Hydra* (Dupre et al., 2017 and Badhiwala et al., 2018):

[see Experiment E5 above for more details]

4. While I appreciate the resection studies to get at "loss-of-function" experiments, this is a somewhat blunt hammer approach, and potentially confounds clear interpretation. It is important to establish which neurons are killed and to what extent, and how many, if any began to regenerate throughout this process. The concern is that residual activity from neurons not ablated may be driving this response, especially in light of the authors surprising finding that "footless" animals show that the aboral nerve ring is not required for spontaneous or mechanosensory responses.

We thank the reviewer for the suggestion to establish which neurons and to what extent they are removed with our resections. To address this point, we performed qPCR on different resectioned *Hydra* (Experiment E3, see above) to identify which neuron subtypes are removed and to quantify to what extent, and how many, might regenerate. Based on the time to experiment post-resections, we expect little neuronal regenerative activity to have occurred and we confirmed this by measuring expression of neuron subtype biomarkers at 12 hours post-resections (our experiments were performed between 6-12 hours post resection).

Our findings from the expression analysis indicate complete removal of specific subtypes with our resections. For example, in “footless” animals, we find complete loss of ec5 subtypes even at 12 hours post-resections. In “headless” animals, we find near complete loss of ec2, ec1B, ec3C and ec4 subtypes. In “body column’ resections, we find the near complete loss of neuron subtypes removed in “footless” and “headless” resections.

[see Experiment E3 above for more details]

Reviewer #2:The Hydra, in the phylum cnidaria, is a near microscopic freshwater animal that has recently resurfaced as an attractive model organism in neuroscience due to its optically accessible transparent body, sparsely distributed neural network, and simple behaviors. In this manuscript, Badhiwala and colleagues use calcium imaging of the Hydra neural network, combined with surgical resection and microfluidics pressure stimulation to identify body regions indispensable for mechanosensory activity. They report that while resection of the aboral region did not abolish the mechanical response, resection of the oral region attenuated this response, while combined resection of oral and aboral regions showed the greatest effect. They also find a correlation between reduced stimulated activity and spontaneous activity, suggesting a common mechanism that gives rise to both activities. While this study takes on an innovative approach by using a microfluidics device to mechanically stimulate the hydra under optical recording there are a number of conceptual and technical limitations. Perhaps my biggest concern is that despite the potential, the data are rather low resolution (body transections and bulk calcium responses) and as such the conclusions supported by the provided data do not significantly extend our current knowledge.

We thank the reviewer for recognizing our innovative methodology to work around technical hurdles. We thought carefully about how to make additional use of our data and thank the reviewer for suggesting individual cell analysis. While it is not feasible to track every cell because they lack a nuclear RFP to automate tracking, we were able to manually track several neurons in the oral and aboral ends. By analyzing these data, we have made two new findings that we found surprising:

1. There are a group of neurons whose pattern of activity is qualitatively different from neurons that belong to contraction burst or rhythmic potential networks. We refer to these neurons as ‘mechanically responsive’ (MR) neurons because these neurons showed a clear response following mechanical stimulus but not spontaneous contractions. The CB network showed very similar activity for both stimulated and spontaneous contractions. Together, these data show a unique neural program for simulated vs spontaneous contractions.

2. The MR neurons showed a significantly delayed response to mechanical stimulation responding approximately 10 seconds after a stimulus evoked contraction. These data suggest that these MR neurons (found primarily in the oral network) may have a separate activation pathway that parallels the activation of the CB neurons and plays a unique role in stimulated contractions compared to spontaneous contractions. This is the first evidence of parallel sensory processing in *Hydra* and was not expected from our initial data.

We have included these new results and a discussion of how they advance our knowledge on pgs. 15-17. We thank the reviewers for suggesting we take this closer look – we were surprised by what we found!

1. The authors have designed a microfluidic device that allows them to simultaneously mechanically stimulate, monitor movement and functionally image a hydra. The highly quantifiable nature of the microfluidic device a great asset, although this potential is not deeply explored. While I can see how the microfluidic stimulation could offer benefits over fluid jet or blunt probe, more in-depth characterization is needed.

We thank the reviewer for this suggestion and have performed new experiments and analysis (Analysis A1, see above) to further characterize this mechanical stimulation technique as described in the response to point 2 below.

2. What is the spatial distribution of the pressure pulse stimulus on the Hydra body? How far does the mechanical force spread from the region directly touching the pressure valve?

We performed additional experiments and analysis (Analysis A1, see above) to map the distribution of mechanical forces from the microfluidic valves. We found the majority of the force is localized to cells and tissue bordering the valve (~ 250 μm around the valve).

The microfluidic valves are 400um in diameter and thus pressure is directly applied in that region. However, the perceived stimulus is likely a combination of stress and strain. As the tissue directly under the valve is compressed, the gelatinous *Hydra* tissue along the bordering the value is pushed/stretched away from the valve center. This indirect force applied on surrounding tissue from elastic deformation of ~400 μm diameter patch of tissue may be non-trivial to quantify. We used displacement of individual neurons throughout the body to indirectly quantify the effects of the mechanical stimulation.

[see Analysis A1 above for more details.]

3. The use of the microfluidic device was limited. Have the authors attempted to map mechanical sensitivity across the Hydra body by stimulating different sites?

We agree with the reviewer that it is important to test multiple areas across the *Hydra* body to characterize mechanical sensitivity and thank the reviewer for the suggested experiment. We found the aboral region of the *Hydra* is less sensitive to mechanical stimulation than the center of the body (Experiment E1, see above).

[see Experiment E1 above for more details]

4. The authors spatially average a population response from a large region of interest as compared to recording calcium responses from single neurons. This should be specifically stated in the Results section. More importantly, to provide insight into network function much smaller ROIs over multiple sites are needed instead of the bulk activity of the entire peduncle. This analysis would allow the authors to significantly extend the findings from this work, as the lure of the optically clear and small hydra is that neural representation and coding can be tracked over large portions of the network at cellular resolution.

We completely agree that to provide insights into the network function we need small ROIs/ individual cell tracking. Although there are technical limitations to this for our organisms (highly deformable body makes it difficult to track individual neurons using calcium fluorescence), we manually tracked several neurons in the oral and aboral regions (Experiment E2, see above) and found at least two networks of neurons that differently coordinate the spontaneous and stimulated behaviors.

[See Experiment E2 above for more details]

We added this new text to the main text to explain the use of large region of interest:

“We also found that the calcium-sensitive fluorescence averaged over a region of interest (ROI) surrounding the peduncle faithfully represented the contraction pulses and bursts measured from individual neurons (Figure 2 —figure supplement 2). […] Based on these experiments, we define Hydra’s “mechanosensory response” as calcium spikes in neural activity from the peduncle ROI and the associated calcium spikes in the epithelial muscles from the whole body if they occur within one second of mechanical stimulation onset (Figure 2c, d and Figure 2 —figure supplement 6).”

To show the highly synchronous nature of peduncle nerve ring neurons that allows us to use a large peduncle ROI when measuring the neuronal response (contraction bursts or pulses) to increase the throughput of our analysis, we performed an additional short experiment and analysis.

We describe this new experiment with new text in our methods:

“Average calcium fluorescence from large and small ROIs in the peduncle

We performed fluorescence imaging in transgenic Hydra expressing GCaMP6s in neurons for ~ 1 min during spontaneous behaviors (Hydra was not stimulated). […] As a result, for all other experiments we use large peduncle ROI when measuring the neuronal activity.”

We added a new supplementary figure (Figure 2 —figure supplement 2).

5. It is unclear where the recorded signals are coming from and if movement is creating artifacts. Have authors corrected for movement? The supplemental movies show a stationary region of interest and moving animal, in some cases parts of animal moving in and out. Furthermore, please explain how the background is subtracted. There is a large fluorescent signal coming from of the entire body/ middle columnar part of the body and spontaneous firing that makes interpretation of the data difficult.

To further validate that the signal captured from our ROIs represent neuronal calcium activity and not motion artifacts, we compared measurements from pan-neuronal GCaMP expressing animals to pan-neuronal GFP expressing animals (Experiment E5, see above).

(see Experiment E5 above for more details)

6. Contraction is a behavioral response of the animal; however, the authors use 'contraction' do describe calcium imaging responses throughout the figures and text. This should be avoided.

We thank the reviewer for this suggestion and have revised the text to refer to these calcium spikes as contraction bursts or contraction pulses and define this as the calcium spike associated with contractions (pg. 7, line 153). We also updated this terminology on pgs 8-9 and 15-20.

We modified the text to define calcium imaging responses:

“This nerve ring activity appeared as either a single bright calcium spike (or “contraction pulse”) or a volley of bright calcium spikes (or “contraction burst”). We also found that calcium-sensitive fluorescence averaged over a region of interest (ROI) surrounding the peduncle faithfully represented the contraction pulses and bursts measured from individual neurons (Figure 2 —figure supplement 2).”

“Based on these experiments, we define Hydra’s “mechanosensory response” as calcium spikes in neural activity from the peduncle ROI and the associated calcium spikes in the epithelial muscles from the whole body if they occur within one second of mechanical stimulation onset (Figure 2c, d and Figure 2 —figure supplement 6).”

7. The authors may wish to rephrase the wording of the paper title. I do not think this work establishes definitively that "multiple nerve rings" are important for coordinating mechanosensory behavior.

We thank the reviewers for the suggested experiments that allowed us to make interesting findings! Specifically, we found the oral and aboral neurons have independent activity patterns indicating distinct networks are involved in coordinating spontaneous and stimulated contractions. Based on the results from these additional experiments, we have modified the tile of our paper to: “Multiple neuronal networks coordinate Hydra mechanosensory behavior”.

[See Experiment E2 above for more details]

8. Furthermore, the claim that the observed "linear relationship" between the spontaneous contraction probability and resection type is evidence for shared neural pathways is not strongly supported by the provided data. These data are fairly coarse resolution and include only 3 animals in each group with highly variable responses (Figure 4C). Additionally, they do not provide evidence to distinguish the motor circuits they hypothesized these neural nets converge upon.

We thank the reviewer for this feedback, and we have replaced this figure with analysis from manually tracking the calcium activity from several neurons which show the independent activities of the oral and aboral networks during stimulated and spontaneous contractions.

[see Experiment E2 and Analysis A2 above for more details]

Reviewer #3:The authors probe mechanosensory processing in Hydra by measuring calcium activity in neurons and muscles in response to precise mechanosensory stimulation in whole and resected animals. The authors' claims are well supported by the evidence. The development of a mechanosensory delivery system for Hydra is also a significant methodological advancement. Taken together, the work advances our understanding of the Hydra nervous system and is a needed step towards developing Hydra as a powerful model for systems neuroscience.

We thank the reviewers for recognizing the significance of our methodological advancement and the significance of our key findings.

1) One issue is that different measures of "mechanosensory response" are used at different places in the manuscript. In some contexts, a response is defined as calcium activity in neurons (Figure 2), and elsewhere as calcium activity in muscles (Figure 3 and 4). And in Figure 2 SuppFig2 muscle contractions are also measured using MeKs. The relation between neural activity, muscle activity and body movement is of course of high interest, and the paper explores this. But, if technically possible, it would be helpful to report a single metric of behavior that could be used in all experiments. For example, it might be possible to use video of the animal's pose or body length to measure contractions in all experiments. At a minimum the reasoning behind choice of measurement of response for each experiment could be discussed explicitly.

Body length proved to not be a reliable method for measuring the behavioral response in *Hydra* due to the technical limitations of working with calcium fluorescence images (Analysis A3, see above).

[see Analysis A3 above for more details]

As alternatively suggested, we discuss the reasoning behind our choice of measurement of response on pg. 7 (line 160-167).

2) Related: Without a consistent measure of behavior, it will be important to further clarify figures so that a reader can tell at-a-glance how contraction probability is being measured.

We have updated the legends of each figure (Figure 2-4; Figure 2 —figure supplement 5-7 and 9; Figure 3 —figure supplement 3 and 4) to indicate the measure of response.